# PerceptionLM: Open-Access Data and Models for Detailed Visual Understanding

**Jang Hyun Cho**[1,2,*,†], **Andrea Madotto**[1,*], **Effrosyni Mavroudi**[1,*], **Triantafyllos Afouras**[1,*], **Tushar Nagarajan**[1,*], **Muhammad Maaz**[3,*,†], **Yale Song**[1,*], **Tengyu Ma**[1,*], **Shuming Hu**[1,*], **Suyog Jain**[1], **Miguel Martin**[1], **Huiyu Wang**[1], **Hanoona Rasheed**[3,†], **Peize Sun**[1], **Po-Yao Huang**[1], **Daniel Bolya**[1], **Nikhila Ravi**[1], **Shashank Jain**[4], **Tammy Stark**[4], **Shane Moon**[4], **Babak Damavandi**[4], **Vivian Lee**[1], **Andrew Westbury**[1], **Salman Khan**[3], **Philipp Krähenbühl**[2], **Piotr Dollár**[1], **Lorenzo Torresani**[1,*], **Kristen Grauman**[1,2,*], **Christoph Feichtenhofer**[1,*]

[1]Meta    [2]UT Austin    [3]MBZUAI    [4]Meta Reality Labs
[*]Joint first author    [†]Work done during internships at Meta    [*]Project lead

## Abstract

Vision-language models are integral to computer vision research, yet many high-performing models remain closed-source, obscuring their data, design and training recipe. The research community has responded by using distillation from black-box models to label training data, achieving strong benchmark results, at the cost of measurable scientific progress. However, without knowing the details of the teacher model and its data sources, scientific progress remains difficult to measure. In this paper, we study building a Perception Language Model (PLM) in a fully open and reproducible framework for transparent research in image and video understanding. We analyze standard training pipelines without distillation from proprietary models and explore large-scale synthetic data to identify critical data gaps, particularly in detailed video understanding. To bridge these gaps, we release 2.8M human-labeled instances of fine-grained video question-answer pairs and spatio-temporally grounded video captions. Additionally, we introduce PLM–VideoBench, a suite for evaluating challenging video understanding tasks focusing on the ability to reason about "what", "where", "when", and "how" of a video. We make our work fully reproducible by providing data, training recipes, code and models at https://github.com/facebookresearch/perception_models.

## 1 Introduction

Vision-language models (VLMs) are now a key part of computer vision research and are widely used in both academia and industry. Many of the strongest performing VLMs are *closed-source*, meaning their design, training methods, and the data they use are not publicly shared. To stay competitive, the research community has started to catch up to the proprietary models by using a straightforward approach — *distillation from black-box models* [1–5], where proprietary models are directly used to label training data [3, 6, 7], directly leading to strong benchmark results.

Although distillation will unlock strong performance, there are two main issues for basic research. First, it makes it hard to track scientific progress. Specifically, we cannot tell if better results on benchmarks are due to advances in model design or training, or simply because the proprietary *teacher* models were trained on the evaluation sets of widely used benchmarks or *internal data* collected to resemble them — this information is not available. Second, the heavy reliance on distillation leads to a fundamental misunderstanding of the effectiveness of current methods for training VLMs *from scratch*. Several key questions remain unanswered, including the significance of each training stage,

39th Conference on Neural Information Processing Systems (NeurIPS 2025).

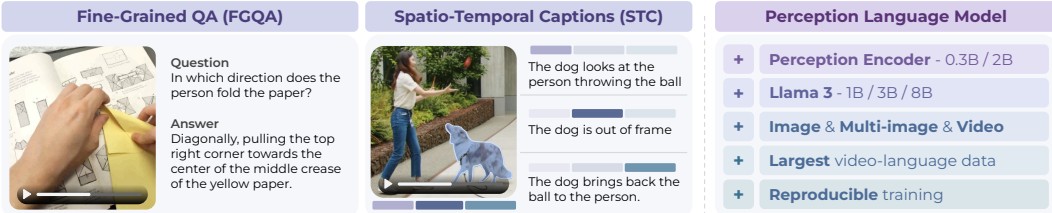

Figure 1: We introduce the largest collection of manually annotated fine-grained activity QA and spatiotemporal captioning data (left panel). Together with this data, we train and release PLM —open and fully reproducible models to facilitate research in vision-language model training (right panel).

the influence of synthetic data , the *data gaps* that the research community should prioritize, and which of these gaps are currently being artificially addressed by distillation from proprietary models.

To better understand these challenges, we develop the Perception Language Model (PLM), a fully open and *reproducible* model for transparent research in image and video understanding (Fig. 1 right). PLM consists of a vision encoder with a small scale (<8B parameters) LLM decoder. We start by an analysis of standard training pipelines with available data, without any proprietary model distillation. We investigate large-scale synthetic data and establish key *scaling laws* to identify critical data gaps that limit *video understanding* performance, especially for *spatio-temporal reasoning* and *fine-grained understanding tasks.*

To fill these gaps, we create 2.8M high-quality human-labeled instances of fine-grained video QA and spatio-temporally grounded video captions, see Fig. 1. This release is nearly an order of magnitude larger than the largest existing video datasets of each type [8, 9]. Our model, dataset and benchmark push the boundaries of video understanding, and provide a foundation for reproducible and transparent training and evaluation of VLM research. Across 40 image and video benchmarks, we achieve comparable performance with existing state-of-the-art open-weight models (*e.g.*, InternVL2.5 [10]), *without* distilling from proprietary models, and greatly outperform fully open models (*i.e.*, Molmo [11]).

## 2 PLM: Overview

In this section, we overview the *model*, *training stages* and *training data* involved in the development of PLM. Please refer to Fig. 8 for a detailed overview and Appendix A for additional details.

**Model.** PLM consists of a vision encoder and language decoder, where a pre-trained Perception Encoder (PE) [12] is connected to the Llama 3 [13] language decoder (1B, 3B, or 8B parameters) with a 2-layer MLP *projector*. We use PE L/14 for Llama3.2 1B and 3B, and PE G/14 for Llama3.1 8B. For image input, PLM incorporates dynamic tiling to support high resolution images for up to 36 tiles of $448^2$ reso-lution, where each tile undergoes $2 \times 2$ average

|  | **Stage 1** Warmup | **Stage 2** Midtraining | **Stage 3** SFT |
|---|---|---|---|
| Modality | Image | Image + Video | Image + Video |
| Data | 1M Synthetic | 72M Mix | 19M Mix |
| Training | Projector | Full | Full |
| Downsampling | - | $2 \times 2$ | $2 \times 2$ |
| Tiles/Frames | 1/- | 16/16 | 36/32 |

Table 1: Summary of three training stages to train PLM. See Appendix Table 8 and Table 9 for data splits.

pooling to compress the visual tokens. For video input, PLM uses 32 frames at $448^2$ resolution, where the same pooling is applied across the spatial dimensions of each video frame.

**Data.** The data used to train the PLM consists of *synthetic* and *human-annotated* samples. Synthetic data enhances the *general* capabilities of PLM, while *human-annotated* data broadens these capabilities to encompass more complex tasks. Synthetic data is sourced from a diverse array of image and video datasets, covering fundamental VLM capabilities such as OCR, chart/document/diagram understanding, image/video captioning, and visual question answering.

We design data engines for each data modality (*e.g.*, natural images, charts, documents, figures, egocentric and exocentric videos) to efficiently scale up, creating ~66.1M samples (§3). The synthetic data can be noisy, but is available at large scale; on the other hand, human-annotated data provides rich, high-quality supervision for image and video tasks. Here, we combine existing human annotations of diverse image and video sources, with our own collected human-annotated data, specifically geared towards *fine-grained video understanding* and *spatio-temporally grounded reasoning* (§4).

**Training stages.** PLM trains in three stages:

**1. Projector warm-up.** First, we freeze the vision encoder and LLM and only train the vision projector on a small amount of synthetic image data. This *warms-up* the newly initialized parameters in the projector and improves stability for later stages. We use $1M$ images from SA-1B [14] with the image captions generated by our data engine (§3).

**2. Large-scale midtraining with synthetic data.** Next, we train PLM on diverse domains of images and videos *at scale*, using a maximum of 16 tiles for images and 16 frames for videos. PLM sees around 64.7M images and videos with synthetically generated captions and question-answer pairs. We employ our data engine to scale up synthetic data generation (see §3).

**3. Supervised fine-tuning with human-annotated data.** Finally, we train PLM with higher image resolutions and more video frames, using up to 36 tiles for images and 32 frames for videos. In this stage, we tackle more challenging video tasks, including *fine-grained QA* and *spatiotemporally grounded reasoning*.

|  | Samples | Type | Stage |
|---|---|---|---|
| *Our Human-annotated (2.87M)* | | | |
| PLM–FGQA | 2.4M | Fine-grained | 3 |
| PLM–STC | 476.2K | R(D)Cap + RTL | 3 |
| *Our Synthetic (66.1M)* | | | |
| Natural Images | 15.9M | Caption | 1,2,3 |
| Charts & Documents | 31.9M | Caption | 2,3 |
| Videos Mix | 17.5M | Mix. | 2,3 |
| Ego4D | 880K | Cap. + QA | 2,3 |
| *Existing Open Source (6.52M)* | | | |
| Image (92 datasets) | 5.6M | Diverse | 2,3 |
| Video (27 datasets) | 920K | Diverse | 2,3 |

Table 2: Summary of the data mix for training PLM. See Table 10 for the full data blend.

Table 1 shows an overview of our training setup for each stage. Appendix A.1 provides the complete training recipe for each stage, including hyperparameters and data sources. The compute cost of each PLM training stage is detailed in Tab. 7 in the Appendix.

## 3   Synthetic Data Generation and Scaling

The predominant paradigm for VLM training is to generate synthetic annotations as cheap alternatives to human-labeled data [1, 10, 11, 15–18]. Although seemingly promising to get the best results on benchmarks, the majority of such data shared in the community is *derived from proprietary models*. This trend makes it hard to decouple scientific progress from proprietary distillation impact. In this section, we explore the efficacy of the current paradigm for VLM training in a *transparent* manner. We design our data engine entirely from *open-source* models and scale the synthetic data generation to around 66.1M samples of images and videos. We establish the scaling laws of training from synthetic data on standard VLM tasks, including image, OCR/document, and video tasks.

### 3.1   Data Engine

Our data engine is designed to target *base* capabilities of VLMs for image and video understanding.

**Image Data Engine.** We generate short and long captions, as well as question-answer pairs, for natural images and those containing documents, diagrams, and text recognizable by optical character recognition (OCR). We prompt openly accessible Llama 3 [13] model to produce factual, detailed image captions while minimizing hallucinations. To create *informative* question-answer pairs, we utilize OCR data, captions, and other *metadata*, which are fed into the prompt of a text-only LLM.

**Video Data Engine.** For videos, we first use an off-the-shelf scene detector [19] to extract video clips of approximately 30 seconds duration. Then, we extract the keyframes and generate frame-level captions using Llama 3, and video captions using our initial PLM trained with Stage 1 and Stage 3 data as shown in Table 2. We then employ an LLM to refine the frame-level and video captions by incorporating existing video metadata (*e.g.*, action labels, time tags) into a cohesive, detailed video-level caption. Similarly, we generate question-answer pairs from the video-level captions.

The resulting synthetic data is large-scale and diverse – 66.1M samples carefully curated from a variety of image and video sources including natural images, in-the-wild text, chart, figures, documents, egocentric and exocentric videos. Additional details are in Appendix I.

## 3.2 Scaling Laws with Synthetic Data

We examine scaling properties of our synthetic data under controlled setup and establish *scaling laws*.

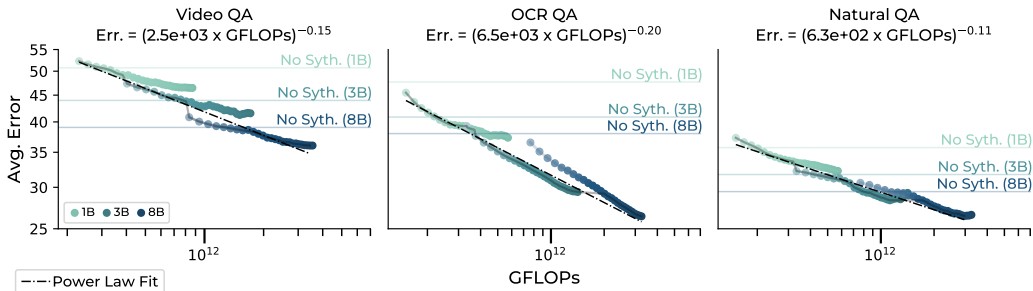

Figure 2: **Synthetic Scaling Plots.** Relationship between Average Error across benchmarks and training compute (in floating-point operations) for various PLM models. We report average errors across Video QA tasks [8, 20–24], OCR QA tasks [25–28], and Natural Images tasks [29–34]. Model's performance using only human-labeled data subset are reported (No Syth.) as well as the actual power-law fit of each subcategory.

**Setup.** To establish power-law relationship between compute and *validation-set errors* of downstream benchmarks, we vary the scale of synthetic data, language model decoders (1B, 3B, and 8B), vision encoders (300M and 2B), and resolution/number of frames. For each configuration, we train a model with the 66.1M synthetic data from our data engine and 6.5M publicly available human-labeled data, following stage 2 training described in §2. At every 2M samples, we evaluate PLM on three categories of downstream benchmarks (*VideoQA*, *OCR QA*, *Natural QA*), constructed from 20 vision-language understanding benchmarks that provide a comprehensive and general evaluation of multi-modal large language models. We compute the *pareto frontier* of these data points and fit a power law relationship: $\texttt{Err.} = (\beta \times \texttt{FLOP})^\alpha$ and compare the exponents $\alpha$ of the power function as *scalability* of each setup, where a smaller $\alpha$ implies better scaling.

**Scaling with decoder size.** Fig. 2 shows the scaling behavior of PLM across various LLM sizes. We show validation-set errors and training compute on a logarithmic scale, with the black linear line representing the power-law relationship between them. Different colors (green, turquoise, and blue) represent different language model scales (1B, 3B, 8B) while keeping the vision encoder size constant at 300M. As described in the setup section above, we show the power law fit of the pareto frontier in each benchmark category. We also show the results of PLM only trained on 4M *human-labeled* datasets as baselines, denoted with horizontal lines of each color. The gap from the horizontal line to the data point marks the impact of the synthetic data. Interestingly, all three categories of benchmarks demonstrate clear power-law relationship between compute and average benchmark errors, with the power law exponent ($\alpha$) of $-0.15$, $-0.20$, and $-0.11$ for Video QA, OCR QA, and Natural Image QA, respectively. In Appendix B, we provide more details and extend the analysis to (1) *scaling the encoder size*, and (2) *scaling the image resolution and video frames*.

**Limitation of synthetic data.** In Fig. 3, we evaluate stage 2 on an extended set of video benchmarks. Specifically, we show the result of 7 *challenging video tasks* on fine-grained activity understanding [35–39], temporal grounding [40] and long-video reasoning [41]. Unlike generic, high-level understanding (*e.g.*, "what is happening in this video"), the "challenging" tasks require a thorough understanding of video in space and time, and fine-grained semantic details. As shown, the challenging video tasks ("HardQA" in lavender, plum, magenta) show a poor scaling trend ($-0.03$) compared to general video QA ($-0.15$). The stark difference between the two power law fits shows that *scaling synthetic data is only effective for established, base tasks*. Extending VLMs to these more challenging, complex tasks still remain unsolved. Next, we address this challenge with high-quality human-annotated video data, **PLM–FGQA** and **PLM–STC**.

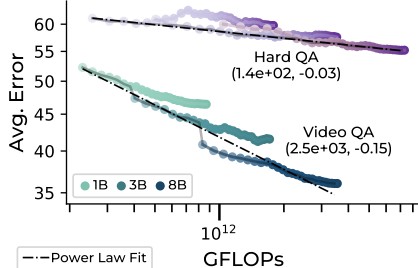

Figure 3: **Limitation of synthetic data.** Challenging video tasks (HardQA [35–41]) do not scale well with synthetic data.

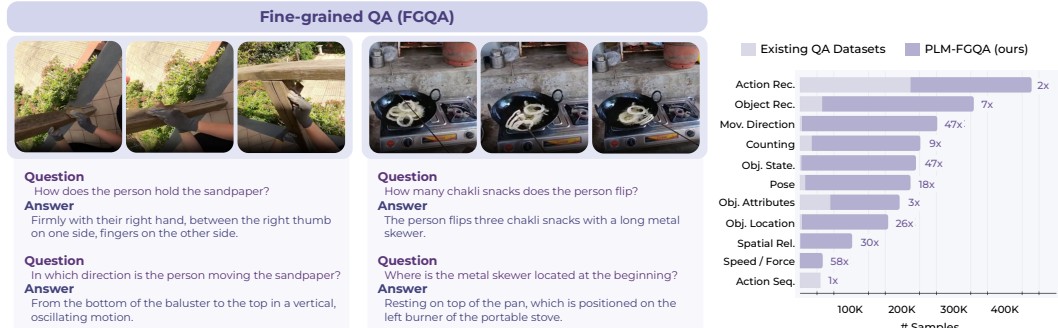

Figure 4: **Overview PLM–FGQA**. Examples of question-answer pairs from PLM–FGQA, focusing on fine-grained human activity understanding. PLM–FGQA is approximately 8 times larger than the largest existing human-annotated video QA dataset and addresses a wide range of fine-grained question types that are scarce in existing video QA datasets, such as ones that cover *direction of movement*, *object states*, *locations* and *spatial relations*.

# 4   Human-annotated High Quality Data

As shown in Fig. 3, the current paradigm with synthetic data has run out of steam. Training from tens of millions of synthetically annotated data hardly improves our model on new, *challenging* video benchmarks. Beyond standard VLM tasks, these benchmarks focus on advanced capabilities such as fine-grained activity understanding, temporal grounding, and long video understanding. Perhaps, the knowledge that these benchmarks examine is simply not present in the initial training set of our data engine nor in existing human-annotated data. Our community lacks high quality datasets for detailed visual understanding to start from, that covers *what*, *where*, *when*, and *how* of activities in video. To address this gap, we introduce two large-scale, human-annotated video datasets:

**PLM–FGQA** is a fine-grained video QA dataset collected by asking human annotators to watch a short video segment and answer model-generated questions which focus on "*what*" activities humans perform and "*how*" they perform these activities. Question types include fine-grained recognition (action and object), fine-grained temporal perception (direction of movements, repetition counts, hand pose etc.), and fine-grained spatial understanding (object locations and spatial relationships). We use a multi-stage data engine to first extract video segments with salient actions from untrimmed videos through temporal clustering and shot-detection. Next, we generate questions and answers using either a text-only LLM or an early version of PLM. Finally, we refine the answers by asking humans to verify or replace them if they are incorrect, resulting in a high-quality QA pairs.

Overall, we collect 2.4M question answer pairs from various open-access video datasets [42–47] spanning over 780k unique video clips from diverse domains (*e.g.*, cooking, DIY, carpentry, automotive and bike repair) and viewpoints (egocentric and third-person); refer to Fig. 13 for domain statistics. This is nearly 8 times larger than the size of the largest existing human-annotated video QA dataset in the community [48]. Moreover, as illustrated by the breakdown of question types[1] in Fig. 4 (top-right), PLM–FGQA contains a large number of annotations about fine-grained details that have been largely missing in existing training video QA datasets [24, 49–56]. Please refer to Table 19 for comparison with existing datasets Table 20 for dataset examples and Appendix F for further details.

**PLM–STC** is a spatio-temporal video captioning dataset that offers detailed activity descriptions for each video. It includes timestamps ("*when*") of each activity and focuses on specific subjects identified by a masklet ("*where*"). We employ a two-stage annotation process to improve efficiency in collecting PLM–STC. In the first stage, annotators select interesting objects that exhibit significant motion changes in the video and use SAM 2 [57] to generate initial mask tublets, which they then refine to ensure high-quality spatial-temporal segmentation. For segments where the subject is out of frame, we automatically supplement "out of frame" caption. In the second stage, a separate set of annotators write temporally localized descriptions of the highlighted subject focusing on the *changes in action across time in relation to the whole video*.

---

[1]obtained with LLM-based tagging.

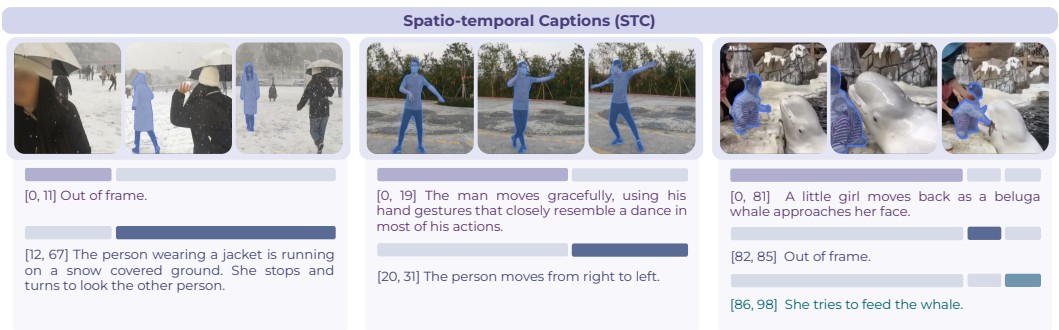

Figure 5: **Overview of PLM–STC**. Examples of spatio-temporally grounded captions from PLM–STC, the first dataset to associate each caption both with a temporal interval as well as a high-fps sequence of segmentation masks of the subject - *i.e.*, masklets (compared to just a temporal interval or a sparse sequence of bounding boxes).

Overall, we collect 194.2K spatio-temporal captions as the first existing large-scale dense video-region captioning dataset. We convert these spatio-temporal captions into three tasks for training: RCap (194.2K): Given the video region and timestamps, the model generates a caption; RTLoc (194.2K): Given the video region and caption, the model localizes the action; and RDCap (122.3K): Given the video region, the model generates dense, localized captions. In total, we construct 194.2K + 194.2K + 122.3K = 522.7K samples, of which 476.2K are used for training and the rest for constructing PLM–VideoBench. Please refer to Fig. 5 for dataset examples, Table 22 for comparison with existing datasets, Table 23 for dataset statistics and Appendix G for further details.

## 4.1 PLM–VideoBench

Our high-quality human-annotated data offers VLMs to train for broader range of capabilities for holistic video understanding. However, existing video benchmarks are not adequately equipped to evaluate these. To this end, we introduce PLM–VideoBench, a novel benchmark focusing on specific activities (what) and their execution details (how) within spatio-temporal contexts (where and when).

**Fine-Grained Question Answering (FGQA).** In this task, a model must answer a multiple-choice question (MCQ) that probes nuanced, fine-grained activity understanding (*e.g.*, painting "vertically" vs. "horizontally" in Fig. 6, first). We report multi-binary accuracy (MBAcc) [39] where each question is split into multiple binary choice questions. Our test set consists of 4,371 question-answer pairs. For more information, including statistics on video clips, segment duration, question types, and benchmark construction, see Table 21 and §F.2.

**Smart Glasses Question Answering (SGQA).** In this task, a model must answer open-ended questions about activities and objects visible in an egocentric video stream recorded by a smart-glasses device (see Fig. 6, second). The questions are designed to simulate real-world scenarios where a user would ask for assistance from their smart glasses. We manually collect the videos using commercially available smart glasses, providing a completely new, unique dataset that reflects modern use-cases such as online AI video assistance and activity coaching. For evaluation, we use LLM-judge accuracy with an open-access model (Llama3.3 70B). The test set consists of 665 human-annotated question-answer pairs. See Appendix H for more details.

**Video Region Captioning (RCap).** In this task, a model must generate a detailed description of an event involving a subject of interest in the video. Given a region masklet and a specified time interval, the model is required to output a caption that accurately describes the event occurring within that interval. Compared to traditional video captioning [47, 58, 59] where the aim is to generate a *video-level* caption, the goal is to generate a *region-level* caption tied to a specific subject (*e.g.*, a person, object or animal) (see Fig. 6, third). The test set contains 10,060 human-annotated instances and we report LLM-judge accuracy with Llama3.3 70B. See Appendix C.3 for details.

**Region Temporal Localization (RTLoc).** In this task, a model must identify the precise time interval within the video when the specified event takes place for the given subject. Given a video, a region masklet and a text description of the event, the model is required to output the start and

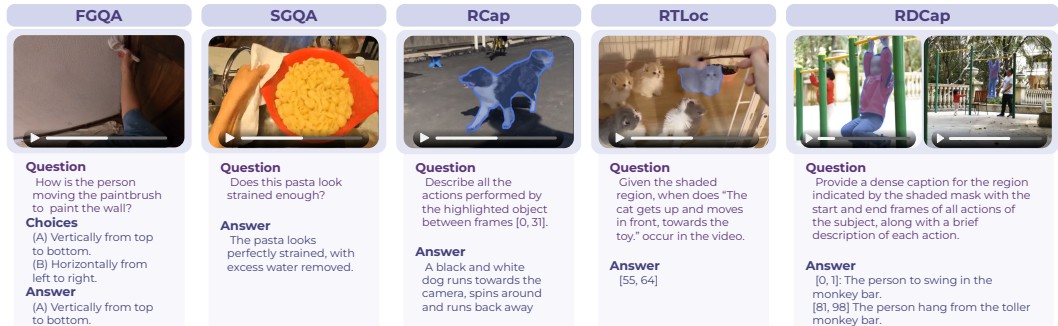

Figure 6: **PLM-Video Dataset** includes fine-grained video QA (FGQA), open-ended QA in videos recorded using smart glasses (SGQA), Spatio-Temporal Captions (STC) post-processed into video region captioning (RCap), video region temporal localization (RTLoc) and video region dense captioning (RDCap) tasks.

end timestamps that correspond to the occurrence of the event (see Fig. 6 fourth). Notably, this task is the inverse of RCap — instead of generating the caption, the model receives it as input and generates the corresponding time interval. We filter the test set to include only the captions that are unambiguously localized, *i.e.*, they map to a single time window in the video. As a result, the test set size is reduced to 7,910 instances compared to RCap. We report average recall@1 over IoU thresholds $(0.3, 0.5, 0.7, 0.9)$. See Appendix C.3 for details.

**Region Dense Video Captioning (RDCap).** In this task, a model must generate a detailed description of all events involving a specific subject of interest (*e.g.*, person, animal, or object) in a video. Given a video and a region masklet, the model must produce a sequence of (`start, end, caption`) tuples that cover the entire duration of the video, including periods when the subject is not visible (see Fig. 6, last). This task is a composition of RTLoc and RCap, requiring the model to produce both temporal windows for events as well as captions directly from the video. The test set contains 2,620 samples and we report the SODA score [60] which uses an LLM judge. See Appendix C.3 for details.

## 5 Experiments

We first overview the baselines and evaluation setting (§5.1). We then compare benchmark results of PLMs with the baselines on a broad collection of image (§5.2) and video (§5.3) tasks as well as on our PLM-VideoBench (§5.4). Finally, we provide analyses on data and model ablations (§5.5).

### 5.1 Setup

We compare PLMs against the following two classes of baselines:

- **Proprietary models** such as GPT-4o [61] (`gpt-4o-2024-11-20`), Gemini-Pro 1.5 [62] and Gemini-Flash 2.0 [63]. We use API calls to evaluate these models.
- **Open-access models** such as Molmo-O [11], LLaVA-OneVision [64], Qwen2.5-VL [15] and InternVL2.5 [10] — state-of-the-art *open-access* models, for which model scale, architecture and inference code are available. We use the official inference code for all models.

**Inference protocol.** For mask inputs in PLM–VideoBench, we overlay a colored box on the video frames to specify the regions. We report validation set performance unless specified (in brackets) under the benchmark name. Metrics marked with † use LLM as a judge. Complete implementation details including inference hyper-parameters, task prompts, judge prompts and proprietary model evaluation protocol can be found in Appendix C.4.

### 5.2 Image Benchmark Results

We evaluate PLM on a total of 20 image benchmarks. **Charts, Diagrams and Documents**: answer questions that require parsing images of documents and diagrams; **Image Captioning**: generate a short/detailed caption, **Perception and Reasoning**: answer questions of varying difficulty about objects, actions, functional correspondence, multi-view reasoning, spatial layout etc. and **Hallucination**: evaluate robustness to *hallucinated* details. More details are in Appendix C.1.

Table 3:

| Model | Charts, Diagrams and Documents | | | | | | Perception and Reasoning | | | | | Hard Perception | | | Halluc. | |
|---|---|---|---|---|---|---|---|---|---|---|---|---|---|---|---|---|
| | DocVQA (test) acc [26] | ChartQA acc [65] | TextVQA acc [66] | InfoQA (test) acc [27] | AI2D (w/o mask) acc [67] | OCRBench acc [28] | MMMU (val) acc [68] | VQAv2 (val) acc [31] | OK-VQA acc [69] | VizWiz acc [33] | SEED (image) acc [70] | BLINK (multi-image) acc [71] | CV-Bench acc [72] | RealWorldQA acc [29] | VSR acc [73] | POPE acc [32] |
| GPT-4o [61] | 92.8* | 85.7* | 75.3 | 80.7* | 94.2* | 810 | 70.7* | - | 63.9 | - | 77.1* | 68.0* | 72.5 | 73.9 | 78.0 | 87.2* |
| Gemini 1.5 Pro [63] | 94.0 | 84.2 | 74.8 | 81.0* | 95.7 | 830 | 63.2 | - | 63.9 | - | 77.8 | 59.8 | 81.0 | 66.3 | 76.1 | 88.2* |
| Gemini 2.0 Flash [63] | 93.0 | 84.8 | 80.2 | 81.0 | 94.0 | 792 | 69.9* | - | 57.8 | - | 77.0 | 64.4 | 82.3 | 71.9 | 74.8 | - |
| **1B scale** | | | | | | | | | | | | | | | | |
| Qwen2VL-2B [16] | 90.1* | 75.3 | 80.3 | **65.5*** | 84.6* | **809*** | 41.1* | 80.0 | 59.7 | **67.4** | 72.9 | 44.4* | 17.3 | 62.6* | 73.0 | 87.2 |
| InternVL2.5-1B [10] | 84.8* | 75.9* | 72.0* | 56.0* | 77.8* | 785* | 40.9* | 72.2 | 51.5 | 47.4 | 71.3 | 42.4 | 42.1 | 58.3 | 65.4 | **90.2** |
| PLM-1B | **90.7** | **78.6** | **82.1** | 63.0 | **84.9** | 807 | 34.8 | **81.7** | **61.0** | 59.7 | **76.3** | **46.8** | **73.8** | **67.1** | **68.8** | 88.4 |
| **3B scale** | | | | | | | | | | | | | | | | |
| Qwen2.5 VL-3B [15] | **93.9*** | 83.1 | 79.3* | **77.1*** | 90.2 | 797* | 53.1* | 80.8 | 63.2 | **71.9** | 73.1 | 47.6* | 54.4 | 65.4* | 78.5 | 88.2 |
| InternVL2.5-4B [10] | 91.6* | 84.0* | 79.3 | 72.1* | 90.5* | 828* | **52.3*** | 80.9 | 64.0 | 61.8 | 75.6 | 50.8* | 55.9 | 64.6 | 80.0 | **91.0** |
| PLM-3B | 93.8 | **84.3** | **84.3** | 74.6 | **90.9** | **830** | 41.2 | **84.3** | **66.8** | 64.0 | **78.5** | **55.4** | **81.4** | **72.4** | **80.4** | 88.7 |
| **8B scale** | | | | | | | | | | | | | | | | |
| Molmo-7B-O [11] | 90.8* | 80.4* | 80.4* | 70.0* | 90.7* | - | 39.3* | 85.3* | - | - | - | - | - | 67.5* | - | - |
| LLaVA-OV-7B [64] | 86.7 | 80.0 | 77.3 | 68.8 | 90.1 | 656 | 48.9 | 83.5 | **69.6** | 63.4 | 76.4 | 49.4 | 75.0 | 66.7 | 78.1 | 89.2 |
| Qwen2.5VL-7B [15] | **95.7*** | **87.3*** | 84.9* | **82.6*** | **93.0** | 864* | **58.6*** | 70.1 | 61.0 | 73.5 | 73.2 | **56.4*** | 11.9 | 69.8 | 80.3 | 87.2 |
| InternVL2.5-8B [10] | 93.0* | 84.8* | 79.3 | 77.6* | 92.8* | 823 | 56.0* | 80.6 | 69.2 | 64.3 | 77.6 | 54.8* | 53.9 | 70.1* | 80.0 | **90.6*** |
| PLM-8B | 94.6 | 85.5 | **86.5** | 80.9 | 92.7 | **870** | 46.1 | **85.6** | **69.6** | 67.0 | **79.3** | 56.0 | **81.3** | **75.0** | **82.8** | 89.9 |

Table 3: **Image benchmarks.** PLM versus proprietary models and open-access baselines of comparable scale. Cells with * are reported numbers from literature, and the remaining are reproduced using official code.

| Model | VCap. | Video QA | | | | | | | Fine-grained Video QA | | | | | T.Loc. | Halluc. | |
|---|---|---|---|---|---|---|---|---|---|---|---|---|---|---|---|---|
| | DREAM-1K F1† [74] | MVBench acc [23] | NExT-QA acc [50] | PerceptionTest (test) acc [24] | STAR acc [21] | Video-MME acc [20] | ActivityNet-QA acc† [51] | EgoSchema (test) acc [22] | TemporalBench MBAcc [39] | TOMATO acc [36] | MotionBench (dev) acc [38] | TempCompass (MCQ) acc [75] | CG-Bench (clue) acc [35] | Charades-STA mIOU [40] | VideoHallucer overall acc [76] | EventHallusion (binary) acc [37] |
| **Proprietary** | | | | | | | | | | | | | | | | |
| GPT-4o [61] | - | 64.6* | 79.1 | - | 70.4 | 71.9* | - | 72.2* | 38.5* | 37.7* | 55.9 | 74.5 | 58.3* | 38.6 | 56.4 | 91.9* |
| Gemini 1.5 Pro [63] | - | 60.5* | 81.6 | 65.9 | - | 75.0* | 56.7* | 71.2* | 34.7 | 32.0 | 56.1 | 75.6 | 50.1* | 34.2 | 56.0 | 80.9 |
| Gemini 2.0 Flash [63] | - | 60.7 | 81.9 | - | - | 70.3* | - | 71.5* | 27.6 | 32.8 | 56.1 | 76.9 | 47.0* | 29.8 | 60.1 | 81.6 |
| **1B scale** | | | | | | | | | | | | | | | | |
| Qwen2VL-2B [16] | 26.8 | 63.2* | 76.4 | 53.9* | 67.3 | **55.6*** | 38.4 | 27.0 | 13.1 | 25.7 | 46.9 | 62.3 | 42.8 | 0.3 | 34.9 | 59.9 |
| InternVL2.5-1B [10] | 27.7 | 64.8 | 74.3 | 59.4 | 73.0 | 50.3* | 60.7 | 55.7 | **27.7** | 25.0 | 45.0 | 56.4 | 40.9 | 0.8 | 31.0 | 38.9 |
| PLM-1B | **34.3** | **70.1** | **80.3** | **72.7** | **83.7** | 49.2 | **62.5** | **60.4** | 18.2 | **25.5** | **52.2** | **64.6** | **43.6** | **55.2** | **49.2** | **79.5** |
| **3B scale** | | | | | | | | | | | | | | | | |
| Qwen2.5 VL-3B [15] | 20.3 | 67.0 | 76.8 | 66.9* | 63.0 | 61.5* | 59.2 | 64.8* | 17.2 | 23.5 | 49.2 | 63.0 | 45.7 | 38.8* | 45.2 | 53.5 |
| InternVL2.5-4B [10] | 29.2 | 71.7 | 82.5 | 67.9 | 77.2 | **62.3*** | 64.1 | 66.6 | **23.7** | 27.4 | 52.7 | 65.2 | **52.0** | 8.4 | 49.6 | 66.3 |
| PLM-3B | **37.4** | **74.7** | **83.4** | **79.3** | **84.8** | 54.9 | **66.2** | **66.9** | 23.4 | **30.9** | **60.4** | **69.3** | 47.2 | **57.7** | **55.5** | **76.5** |
| **8B scale** | | | | | | | | | | | | | | | | |
| LLaVA-OV-7B [64] | 28.0 | 57.1 | 81.0 | 58.1 | 66.0 | 57.7 | 60.5 | 45.4 | 19.5 | 27.6 | 53.7 | 67.8 | 41.2 | 12.1 | 34.7 | 61.1 |
| Qwen2.5VL-7B [15] | 23.3 | 69.6* | 80.0 | 70.5* | 68.1 | **65.5*** | 63.7 | 65.0* | 24.5 | 24.6 | 51.1 | 71.7* | 49.8 | 43.6* | 50.1 | 61.1 |
| InternVL2.5-8B [10] | 28.5 | 72.6 | **85.5** | 68.9* | 77.6 | 64.2* | 66.1 | 66.2* | 24.3 | 29.4 | 53.5 | 68.3* | **53.1** | 14.3 | 57.1 | 60.2 |
| PLM-8B | **35.9** | **77.1** | 84.1 | **82.7** | **84.9** | 58.3 | **67.3** | **68.8** | 28.3 | **33.2** | **61.4** | **72.7** | 46.4 | **58.6** | **57.7** | **77.3** |

Table 4: **Video benchmark results.** PLM versus proprietary models and open-access baselines of comparable scale. Cells with * are reported numbers from literature and the remaining are reproduced using official code.

Table 3 shows our results. Overall, PLM shows strong performance on a wide spectrum of image benchmarks with *solely from open-access data with a white-box data engine*. Additionally, we report Image Grounding task results on RefCOCO/+/g [77] datasets in Appendix Table 15, and show that PLM outperforms both specialist models as well as the VLM baselines in all model scales.

## 5.3 Video Benchmark Results

We evaluate PLM on a total of 25 video benchmarks. We divide these into the following categories. **Video Captioning**: generate a short caption for a video, or a dense description of all events; **Short video QA**: answer a question about a short video (few seconds to a minute), either by selecting from a list of options, or providing a free-form answer; **Long video QA**: answer a question as before, about a much longer video (minutes to hours); **Fine-grained QA**: answer detailed questions about spatial location, motion, temporal information etc.; and **Hallucination**: evaluate the robustness of video models to *hallucinated* details about objects and events.

Table 4 shows video captioning, video QA, fine-grained video QA, and video hallucination results. We achieve strong results on widely adopted benchmarks, despite only using open-access data mix free from proprietary model artifacts, outperforming both the open-access and proprietary models.

Further, we achieve competitive performance on the majority of challenging benchmarks, such as EgoSchema (68.8 %), MotionBench (61.4 %), TOMATO (33.2 %), TempCompass (72.7 %), TemporalBench (28.3 &), Charades-STA (58.6 %), and more. All our model scales show strong performance against both proprietary models as well as open-access baselines of same scale.

Lastly, we also show that PLMs at all scale greatly outperform existing approaches on captioning tasks and hallucination detection tasks, owing to our focus on detailed, fine-grained spatio-temporal annotations in our human-annotated data collection.

## 5.4 PLM-VideoBench Results

We report the result on our proposed benchmark PLM-VideoBench from §4.1 in Table 5. We evaluate our PLM as well as (proprietary and open-access) baselines. In addition, we provide human performance of each subtask in the first row. The results show a significant gap between the baselines and PLM. Proprietary baselines and open-source baselines alike perform reasonably on FGQA tasks, though still 6.5 points lower than PLM (61.2 vs 67.7).

On SGQA, where the video sources and the question-answer pairs are unseen to all models, PLM performs reasonably well, yet 2.1 points short from open-access best (InternVL2.5) and far from the best proprietary model (GPT-4o).

On spatio-temporal tasks (RDCap, DCap, RT-Loc), open source baselines are unable to perform grounded reasoning and default to repeating the same caption for every time interval. Proprietary models perform reasonably well, yet far from the human performance. In all sub-tasks of PLM-VideoBench, PLM shows competitive performance compared to proprietary and open-access baselines. Results for all model scales are in Appendix E.1.

Tab. 18 in the Appendix reports PLM's performance when trained without any human-annotated in-distribution samples. Even in this setting, PLM outperforms Qwen2.5-VL 3B and InternVL2.5 4B on PLM-VideoBench.

| Model | FGQA *MBAcc* | SGQA *acc†* | RDCap *SODA†* | RCap *score†* | RTLoc *meanR* | Avg. |
|---|---|---|---|---|---|---|
| Human perf. | 90.9 | 67.9 | 66.6 | 53.9 | 67.8 | 73.9 |
| **Proprietary** | | | | | | |
| GPT-4o [61] | 61.2 | 63.7 | 20.9 | 35.7 | 33.1 | 51.6 |
| Gemini 1.5 Pro [63] | 57.1 | 49.9 | 14.4 | 33.1 | 27.6 | 44.0 |
| Gemini 2.0 Flash [63] | 58.7 | 44.8 | 13.2 | 30.9 | 27.6 | 42.5 |
| **Open-access** | | | | | | |
| LLaVA-OV-7B [64] | 40.2 | 41.5 | 4.7 | 24.4 | 13.9 | 32.0 |
| Qwen2VL-7B [16] | 49.2 | 44.5 | 4.1 | 17.6 | 15.1 | 35.3 |
| Qwen2.5VL-7B [15] | 49.8 | 43.0 | 2.5 | 21.5 | 10.7 | 34.8 |
| InternVL2-8B [10] | 47.7 | 45.9 | 1.2 | 21.5 | 11.6 | 35.0 |
| InternVL2.5-8B [10] | 53.7 | **48.3** | 5.7 | 26.1 | 8.8 | 38.5 |
| PLM-8B | **67.7** | 46.2 | **52.8** | **46.6** | **59.1** | **55.6** |

Table 5: **PLM-VideoBench results.** We evaluate PLM against baselines and report breakdowns. We report human performance in the first row.

Note that the human performance varies based on the nature of the task and evaluation metrics. For example, FGQA human scores are naturally higher than RCap because the task is structured (select the correct option vs. open-ended) and the metric is objective (accuracy vs. LLM-judge accuracy).

## 5.5 Ablation Studies

**Setup.** We perform an ablation study to assess the importance of each of our proposed data, both synthetic and human-annotated. We start with PLM 3B after stage 2 training, and finetune on 4M short image and video SFT data mix [2] for the data ablation. We evaluate and report average video benchmark performance across five categories — video captioning, short video QA, fine-grained QA, and video hallucination, as well as spatial and temporal tasks, PLM-VideoBench and three image categories — image OCR, image captioning, and image perception. Full details are in Appendix A.3.

**Discussion.** First, we observe that stage 2 synthetic data training boosts model performance across the board. Moreover, adding our PLM-STC data further improves a variety of benchmarks, including PLM-STC (+27.4 points), video captioning (+2.4 points), and most importantly, spatial and temporal tasks (+6.8 points). Adding our PLM-FGQA data improves a distinct set of categories for fine-grained activity understanding; PLM-FGQA (+13.1 points), PLM-SGQA (+7.3 points), Fine-grained video tasks (+1.3 points), video hallucination tasks (+3.0 points), and spatial and temporal tasks (+2.2 points). Using our human-annotated data altogether results in the best performance overall. Further in Fig.7, we show that our human-annotated data improves upon HardQA [35–41], effectively addressing the limitations of synthetic data discussed in §3.2.

---

[2] 3.8M datamix: TextQA 500K, Image QA 2.8M, and Video QA 500K. Each detail can be found in Tab. 10.

Finally, we conduct an ablation study (Tab. 17 in Appendix), where Stage 2 and 3 are merged into a single training phase using a combined data blend under the Stage 3 setup (36 tiles, 32 frames). Despite nearly doubling the total training FLOPs, this unified setting yields worse results than the three-stage pipeline—dropping 1.2 points on image benchmarks and 2.0 points on video benchmarks.

| PLM-Synth. | PLM-STC | PLM-FGQA | Total Average | PLM-FGQA *MBAcc* | PLM-SCQA *acc†* | PLM-ST *3 metric avg.* | Fine-Grained QA *5 benchmark avg.* | Video Cap. *Dream 1K* | Video QA *5 benchmark avg.* | Video Hallu. *2 benchmark avg.* | Spatio&Temp. *4 benchmark avg.* | Image OCR *6 benchmark avg.* | Image Cap. *3 benchmark avg.* | Image Rec. *5 benchmark avg.* |
|---|---|---|---|---|---|---|---|---|---|---|---|---|---|---|
| ✗ | ✗ | ✗ | 48.5 | 39.7 | 34.4 | 6.6 | 42.2 | 24.0 | 67.5 | 64.9 | 50.6 | 76.0 | 64.3 | 63.3 |
| ✓ | ✗ | ✗ | 54.3 | 49.8 | 35.9 | 14.7 | 48.8 | 29.9 | 73.2 | 73.3 | 56.1 | **84.0** | 65.9 | 65.5 |
| ✓ | ✓ | ✗ | 57.9 | 49.9 | 36.2 | 42.1 | 48.6 | 32.3 | 73.9 | 74.2 | 62.9 | 83.8 | 67.5 | 65.0 |
| ✓ | ✗ | ✓ | 56.7 | 62.9 | 43.2 | 15.2 | 50.1 | 30.4 | 74.1 | **76.3** | 58.3 | 83.7 | 64.0 | **65.6** |
| ✓ | ✓ | ✓ | **61.2** | **63.6** | **44.0** | 42.2 | **50.2** | **34.3** | **74.6** | **76.3** | **64.3** | 83.7 | **74.2** | 65.4 |

Table 6: **Ablation.** We show the impact of individual data components in PLM training. For this ablation, we use a reduced the SFT datamix consists of 4M open-access image and video data. Results are aggregated validation-set performance over selected benchmarks in each category of tasks, details in Appendix A.3.

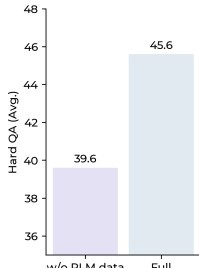

Figure 7: HardQA improves with PLM data.

# 6 Related Work

**Vision-Language Models.** Building on the strengths of large language models (LLMs), several vision-language models (VLMs) have recently been proposed for image understanding [1, 13, 18, 72, 78–82], video understanding [52, 83–89] and joint understanding of both images and videos [10, 16, 64, 90]. These works employ several modeling advancements such as dynamic high resolution inputs [78], adaptive token compression [87, 91], and multimodal positional embeddings [16].

**Open source, open data VLMs.** Training data is a key component in developing powerful VLMs. Many existing approaches train on proprietary data that is not released to the community [61–63, 92, 93] or on data generated using proprietary models (*e.g.*, GPT4o) [3], effectively distilling the *closed* models. Doing so make measuring scientific progress difficult and limits research on how to train VLMs ground-up. Molmo [11] proposes a class of open-data models, however, they are image VLMs trained on relatively small-scale data, limiting their performance as our experiments will show.

**VLM Benchmarks.** Several benchmarks have been proposed to assess the capabilities of VLMs. Popular image benchmarks cover broad perception and reasoning [29, 33, 68, 69, 71, 72, 94–100] as well as capabilities like image captioning [101–103], document/diagram understanding [26–28, 65–67, 70, 104–106], mathematical reasoning [107–109], visual grounding [77, 110] and hallucination [32, 111]. Popular video benchmarks cover video question answering [8, 20, 21, 23, 24, 50–52, 84, 112–117], video captioning [47, 59, 74, 118–121], and hallucination in videos [37, 76]. Many of these video benchmarks remain *image-centric* — they have questions that can be answered with a few frames. Video-centric reasoning in benchmarks has been relatively neglected with benchmarks proposed only recently for long video understanding [22, 35, 41, 48, 122–126] and fine-grained, temporal reasoning [36, 38, 39, 75, 127]. We introduce PLM–VideoBench— a benchmark suite aimed at the core, video-centric capabilities that current benchmarks neglect, namely fine-grained activity understanding and spatio-temporally grounded reasoning.

# 7 Conclusion

This work presents Perception Language Model (PLM), a fully-reproducible vision-language model to transparently tackle visual perception tasks without distillation of private black-box models. We trained PLM using data from existing open-access datasets and synthetic samples generated by our data engine. We identified gaps in detailed video understanding capabilities that cannot be filled with synthetic data. In response, we collected 2.8M human-labels for fine-grained video question answering and spatio-temporally grounded captioning, and created a new benchmark, PLM-VideoBench, to evaluate these capabilities. We hope our open dataset, benchmark, and models will foster transparent research in visual perception.

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

# Appendix

## Table of Contents

# A  PLM Training Details

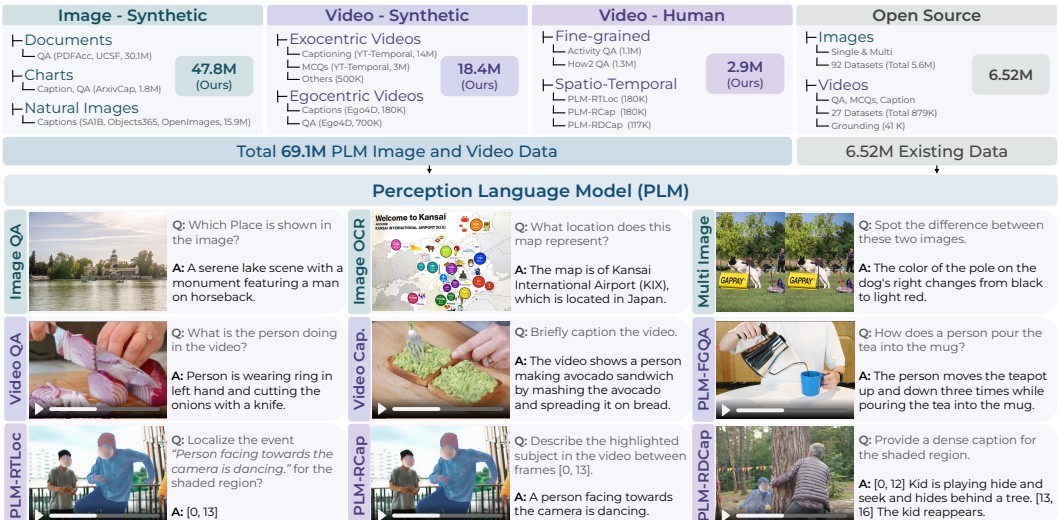

Figure 8: The figure provides an overview of the datasets used in the paper. PLM is trained with *47.8M* synthetic image and *18.4M* synthetic video, and *2.9M* human-labeled video samples. Our data enables PLM to perform a variety of tasks, including standard tasks like Image, Multi-image, and Video QA, as well as *new video tasks* such as Fine-grained QA (FGQA), Region Temporal Localization (RTLoc), Region Captioning (RCap), and Region Detailed Captioning (RDCap).

In this section, we describe the training details of PLM. In §A.1 we describe exact details of training setting such as hyper-parameters and implementation details. In §A.2 we describe our datamix for both synthetically generated and human-annotated parts.

## A.1  PLM Training Setting

For all three stages, we use AdamW optimizer [128] with weight decay of 0.05 and use FSDP [129] with FlashAttention2 [130] for overall implementation based on PyTorch [131].

**Stage 1 training.** In stage 1, we use a subset of SA-1B [14] paired with detailed captions generated by our data engine (§3.1). We use total 1M samples to train PLM with next token prediction loss, with vision encoder and LLM parameters frozen. This stage is commonly known as *warm-up* stage. We use learning rate $1 \times 10^{-4}$ for all model scale with global batch size of 512 and $448 \times 448$ resolution. We use the Perception Encoder [12] L/14 variant for the 1B and 3B PLM models, and the G/14 variant for the 8B PLM model.

**Stage 2 training.** In Stage 2, we train on a total of 72.5M samples. Of these, 66M consist of images and videos with synthetically generated annotations produced by our data engine. The remaining 6.5M samples are a subset of human-annotated images and videos from open-source datasets, which are included in our final datamix described in §A.2. We train with global batch size of 2048, learning rate of $4 \times 10^{-5}$, weight decay of 0.05 for the full set of parameters (vision encoder, projector, and LLM). For both image and video input, we use $448 \times 448$ resolution for each tile/frame, which effectively generate 1024 vision tokens. We apply $2 \times 2$ spatial average pooling to reduce this to 256. We use dynamic tiling with a thumbnail to support any resolution and aspect ratio, similar to prior work [78], and uniform sampling of video frames after preprocessing the videos to 1 fps. We set the maximum number of tiles/frames to be 16, which results in maximum of $(16 + 1) \times 256 = 4352$ and $16 \times 256 = 4096$ vision tokens respectively for images and videos. We train the model with a sequence length of 6144 allowing a maximum of 2048 tokens for the text modality.

**Stage 3 training.** In stage 3, we use total of 19.1M high-quality datamix spanning over multiple image, video, and text modalities. We describe this datamix in §A.2. In this stage, we use global batch size of 1024, learning rate of $1 \times 10^{-5}$ for 8B and $4 \times 10^{-5}$ for 1B and 3B PLM models. We train the full set of parameters for all scales. Similar to stage 2, we adapt dynamic tiling and uniform

frame sampling for up to 36 tiles for image and 32 frames for video, with $2 \times 2$ spatial average pooling, which generates $(36 + 1) \times 256 = 9472$ vision tokens for image and $32 \times 256 = 8192$ vision tokens for video. For all modalities, we use $11264$ maximum training sequence length.

**Training cost.** The full training process for the PLM-3B model, including all three stages, takes approximately 5.5 days on 128 H100 GPUs. The PLM-1B model requires less training time, while the PLM-8B model takes slightly longer. The detailed breakdown is listed in the Table 7 . Further, at inference time, all models can run efficiently on a single GPU: the 1B model on a 24GB GPU, the 3B model on a 40GB GPU, and the 8B model on an 80GB GPU.

| Model | Stage 1 | Stage 2 | Stage 3 |
|---|---|---|---|
| PLM-1B (PE L/14) | 8 GPUs & 3 hours | 128 GPUs & 1.5 Days | 128 GPUs & 2.0 Days |
| PLM-3B (PE L/14) | 8 GPUs & 4 hours | 128 GPUs & 3.0 Days | 128 GPUs & 2.5 Days |
| PLM-8B (PE G/14) | 8 GPUs & 6 hours | 256 GPUs & 3.0 Days | 256 GPUs & 2.5 Days |

Table 7: Training efficiency of PLM models across scales.

## A.2 PLM Training Datamix

Table 10 presents the full data mix used across all training stages apart from our manually collected data in §4. This contains annotations from existing public datasets as well as synthetically generated data (see §3). We filter and include a wide variety of existing datasets spanning across images (captioning, QA, grounding), videos (captioning, QA, temporal localization, region captioning and dense captioning) and text-only datasets to preserve the text-instruction following capabilities of our model. Most importantly, we filter out *every* dataset that contains annotations generated by proprietary models. Table 8 and Table 9 shows the exact number of samples for each datasets in Stage 2 and Stage 3 respectively. Marjory of the data in stage 2 are synthetic, with a focus on captioning samples, since they carry the dense information about the image or video. In stage 3, we have one third of the data, mostly focusing on human annotated samples, covering a large variety of tasks.

## A.3 Ablation Experiment Details

We provide additional details about the ablation experiment in §5.5. We report benchmark average scores across 5 categories, along with the average across all of them. We select a representative set of benchmarks from the full set of image and video benchmarks in §5.2 and §5.3 that report comparable scores so the average results are meaningful. For Video captioning we select Dream 1K and report the LLM-judge score with Llama3.3 70B as judge. for Short Video QA, and Finegrained QA, we select benchmarks that report MCQ accuracy (and exclude open-ended QA). For Hallucination, we include both benchmarks. For Spatial and Temporal tasks, we select BLINK, CVBench, VSR, and Charades-STA. For Image Perception, we choose SEED, MMMU, VQAv2, OK-VQA, and VizWiz. We train the ablation setup of SFT with the exactly matching hyperparameters as our final run; only difference is the size of the SFT datamix.

| Dataset | Num Samples | Type |
|---|---|---|
| *Image Synthetic* | | |
| PDFAcc (QA) [132] | 12M | QA |
| PDFAcc (Cap) [132] | 12M | Cap. |
| UCSF [133] | 6M | QA |
| ArxivCap [134] | 1.8M | Cap./QA |
| SA1B [14] | 10M | Cap. |
| Object365 [135] | 3.5M | Cap. |
| OpenImages [136] | 1.8M | Cap. |
| DocVQA [26] | 50K | QA |
| InfographicVQA [27] | 20K | QA |
| PixmoCap [11] | 600K | Cap |
| *Video Synthetic* | | |
| YT-1B (Cap.) [137] | 14M | Cap. |
| YT-1B (QA) [137] | 3M | MCQA |
| Ego4D (Cap.) [43] | 180K | Cap. |
| Ego4D (QA) [43] | 700K | QA |
| Spoken Moments [138] | 449K | Cap. |
| Charades [139] | 8K | Cap. |
| Kinetics710 [54] | 40K | Cap. |
| DiDeMo [140] | 7.5K | Cap. |
| *Text Synthetic* | | |
| NaturalReasoning [141] | 1M | QA |
| *Human Annotated* | | |
| Image QA [10] | 2.8M | QA |
| Video QA [10] | 570K | QA |
| Video TL [10] | 16K | Temp. Loc. |
| Video Dense Cap. [10] | 10K | Dense Cap. |
| Text QA [10] | 2M | Mix |
| Total | 72.5M | |

Table 8: PLM Stage 2 training data mix.

| Dataset | Num Samples | Type |
|---|---|---|
| *Image Synthetic* | | |
| PDFAcc (QA) [132] | 2M | QA |
| ArxivCap [134] | 1.5M | Cap./QA |
| SA1B [14] | 800K | Cap. |
| Object365 [135] | 300K | Cap. |
| OpenImages [136] | 300K | Cap. |
| DocVQA [26] | 100K | QA |
| InfographicVQA [27] | 50K | QA |
| PixmoCap [11] | 500K | Cap |
| *Video Synthetic* | | |
| YT-1B (QA) [137] | 300K | MCQA |
| Ego4D (Cap.) [43] | 180K | Cap. |
| Ego4D (QA) [43] | 700K | QA |
| Spoken Moments [138] | 449K | Cap. |
| Charades [139] | 8K | Cap. |
| Kinetics710 [54] | 40K | Cap. |
| DiDeMo [140] | 7.5K | Cap. |
| *Text Synthetic* | | |
| NaturalReasoning [141] | 1M | QA |
| *Human Annotated* | | |
| Image QA [10] | 2.8M | QA |
| Image Cap [10] | 36K | QA |
| Image Grnd. [10] | 1.4M | QA |
| Image Misc. [10] | 1.4M | QA |
| Video QA [10] | 570K | QA |
| Video Cap. [10] | 315K | QA |
| Video TL [10] | 16K | TL |
| Video Dense Cap. [10] | 10K | DCap. |
| Video Region Captioning [10] | 15K | Cap. |
| Text QA [10] | 1.5M | Mix |
| *Human Annotated (Our)* | | |
| PLM FGQA | 2.4M | QA |
| PLM STC | 476K | Cap./TL |
| Total | 19.1M | |

Table 9: PLM Stage 3 training data mix.

## Image QA

| The Cauldron | Dataset | Size |
|---|---|---|
| | DVQA [142] | 222222 |
| | PlotQA [143] | 157070 |
| | MapQA [144] | 42761 |
| | OCRVQA [145] | 167646 |
| | Localized Narratives [146] | 199998 |
| | FigureQA [147] | 119999 |
| | Hateful Memes [148] | 9713 |
| | CLEVR [149] | 73181 |
| | CLEVR v1.0 [149] | 70000 |
| | IconQA [150] | 116514 |
| | TextVQA [34] | 21953 |
| | GeomVerse [151] | 11162 |
| | RobuT (wikisql) [152] | 80757 |
| | WebSight [153] | 10000 |
| | Visual7W [154] | 15961 |
| | TallyQA [155] | 100050 |
| | Robut (WTQ) [152] | 42495 |
| | DaTikz [156] | 47974 |
| | CocoQA [157] | 46287 |
| | ChartQA [25] | 27395 |
| | VQAv2 [31] | 82772 |
| | Chart2Text [158] | 35946 |
| | VisText [159] | 35995 |
| | FinQA [160] | 5276 |
| | DocVQA [26] | 12089 |
| | STVQA [161] | 18684 |
| | TAT-QA [162] | 2199 |
| | RenderedText [163] | 10435 |
| | RAVEN [164] | 31418 |
| | IAM [165] | 7549 |
| | A-OKVQA [69] | 17720 |
| | TabMWP [166] | 45439 |
| | CocoQA [157] | 9009 |
| | TextCaps [167] | 21953 |
| | Screen2Words [168] | 16713 |
| | VSR [169] | 2157 |
| | TQA [170] | 9742 |
| | Robut (SQA) [152] | 12769 |
| | VisualMRC [171] | 3027 |
| | ScienceQA [106] | 9947 |
| | VQA-RAD [172] | 313 |
| | InfographicVQA [27] | 2118 |
| | Hitab [173] | 4995 |
| | AI2D [67] | 4863 |
| | Inter-GPS [174] | 2555 |
| | diagram_image_to_text [175] | 595 |
| | MIMIC-IT (CGD) [176] | 70939 |
| | MultiHiertt [177] | 15233 |
| | NLVR2 [178] | 136799 |
| | RAVEN (Multi-image) [164] | 56081 |
| | SpotTheDiff [179] | 19340 |

| | Dataset | Size |
|---|---|---|
| M4 Instruct (Filtered) | STAR [21] | 3032 |
| | NExT-QA [50] | 3870 |
| | VISION [180] | 9900 |
| | FlintstonesSV [181] | 22341 |
| | ImageCoDe [182] | 16594 |
| | VizWiz [33] | 4900 |
| | MIT-States (State Coherence) [183] | 1900 |
| | MIT-States (Prop. Coherence) [183] | 1900 |
| | WebQA [184] | 9338 |
| | Birds-to-Words [185] | 14281 |
| | AESOP [186] | 6915 |
| | RecipeQA (Img. Coherence) [187] | 8699 |
| | CLEVR-Change [188] | 3885 |
| | IEdit [189] | 3456 |
| Ureader (filtered) | ChartQA [25] | 45820 |
| | DocVQA [26] | 69562 |
| | InfographicVQA [27] | 32661 |
| | TextVQA [34] | 69170 |
| | TextCaps [167] | 21324 |
| | VisualMRC [171] | 24456 |
| | WTQ [190] | 16885 |
| | HME100k [191] | 74492 |
| | chrome_writting [163] | 8825 |
| | OK-VQA [30] | 27536 |
| | Geometry3k [174] | 4802 |
| | VQA-RAD [172] | 1793 |
| | **Total** | **2796145** |

### Image Cap.

| Dataset | Size |
|---|---|
| DOCCI [192] | 13362 |
| DCI [193] | 7599 |
| Altogether [194] | 15166 |
| **Total** | **36127** |

### Image Misc.

| Dataset | Size |
|---|---|
| AI2d [67] | 12413 |
| COCO cap. [101] | 414113 |
| GQA-Balanced [195] | 943000 |
| **Total** | **1369526** |

### Grounding

| Dataset | Size |
|---|---|
| VisualGenome [110] | 154792 |
| FLickr Entities [196] | 296332 |
| DCI (Region Caption) [193] | 304912 |
| RefCOCO/g/+ [197] | 212923 |
| VCR [105] | 855577 |
| **Total** | **1398690** |

### Image Synth.

| Dataset | Size |
|---|---|
| DocVQA [26] | 50170 |
| InfographicVQA [27] | 21660 |
| PDFAcc (Cap.) [132] | 12024670 |
| PDFAcc (QA) [132] | 12024670 |
| UCSF [133] | 5953490 |
| ArxivCap [134] | 1859680 |
| SA1B [14] | 9834573 |
| Object365 [135] | 3484584 |
| OpenImages [136] | 1740864 |
| PixmoCap [11] | 584650 |
| **Total** | **47579011** |

### Video QA

| Dataset | Size |
|---|---|
| EgoQA [49] | 7813 |
| NExT-QA (instruct) [50] | 34114 |
| NExT-QA (MCQ) [50] | 34114 |
| PerceptionTest [24] | 2403 |
| ActivityNetQA [51] | 23530 |
| VideoInstruct (human) [52] | 25803 |
| CLEVRER (MC) [53] | 42620 |
| CLEVRER (QA) [53] | 40000 |
| Kinetics710 [54] | 39949 |
| SSv2 (classification) [55] | 40000 |
| VidLN [56] | 43126 |
| VidLN (QA) [56] | 75090 |
| How2QA [8] | 45731 |
| STAR [21] | 35297 |
| Memento [198] | 40060 |
| Memento-MultiImage [198] | 40060 |
| **Total** | **569710** |

### Video Cap.

| Dataset | Size |
|---|---|
| VATEX (en caption) [59] | 259910 |
| Charades (caption) [139] | 11593 |
| ActivityNet (captions) [58] | 33375 |
| YouCook2 [47] | 10337 |
| **Total** | **315215** |

### Video Temporal Loc.

| Dataset | Size |
|---|---|
| HiREST [199] | 7919 |
| Charades [139] | 7566 |
| DiDeMo [140] | 435 |
| **Total** | **15920** |

### Video Region Captioning

| Dataset | Size |
|---|---|
| HC-STVG [200] | 10131 |
| VidLN (UVO subset) [56] | 5296 |
| **Total** | **15427** |

### Video Dense Cap.

| Dataset | Size |
|---|---|
| ActivityNet [58] | 8859 |
| YouCook [47] | 1039 |
| **Total** | **9898** |

### Video Synth.

| Dataset | Size |
|---|---|
| Spoken Moments [138] | 449044 |
| Charades [139] | 7919 |
| Kinetics710 [54] | 39949 |
| DiDeMo [140] | 7566 |
| Ego4D (Cap.) [43] | 183029 |
| Ego4D (QA) [43] | 703935 |
| YT-1B (Cap.) [137] | 14792983 |
| YT-1B (QA) [137] | 3383670 |
| **Total** | **19568095** |

### Text-QA

| Dataset | Size |
|---|---|
| no_robots [201] | 9485 |
| MathQA [202] | 29837 |
| LIMA [203] | 1030 |
| GSM8k (socratic) [204] | 7473 |
| GSM8k [204] | 7473 |
| FLAN [205] | 1386050 |
| Dolly15k [206] | 15011 |
| Magpie Pro (MT) [207] | 300000 |
| Magpie Pro [207] | 300000 |
| **Total** | **2056359** |

Table 10: **PLM training datamix.** Our mix includes synthetic and manually annotated data across a combination of image data (QA, captioning, OCR, Visual grounding), video data (captioning, grounded captioning, dense captioning, temporal localization) and text-only data. Importantly, all data is publicly accessible, and *not* generated by proprietary models.

# B Synthetic Scaling Experiments

In this section we provide additional results to the synthetic scaling experiments in §3.2. We report aggregate benchmark accuracies across three categories — Video QA, OCR QA and Image QA — by selecting representative benchmarks from each category. For VideoQA, these are STAR [21], EgoSchema [22], MVBench [23], VideoMME [20] and PerceptionTest [24]; For OCR QA, these are ChartQA [25], DocVQA [26], InfographicsQA [27], TextVQA [34] and OCRBench [28]; and for Natural Image QA, these are RealworldQA [29], OKVQA [30], VQAv2 [31], and VizWiz [33].

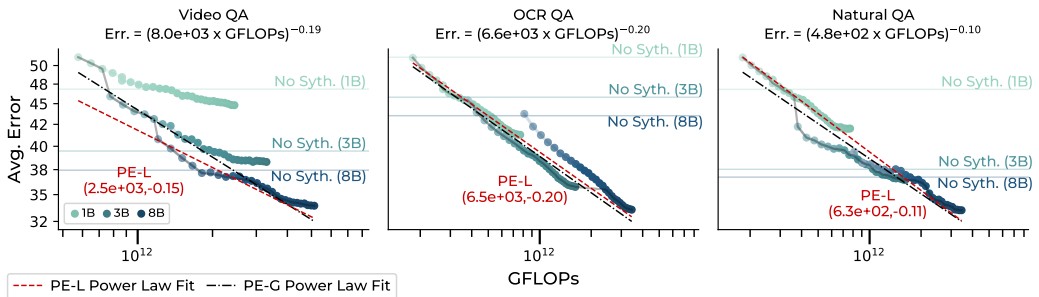

Figure 9: **Scaling with encoder size.** Scaling trends of PE-G vs. PE-L vision encoders. Larger encoders scale better in Video QA tasks while similar scaling in OCR and Natural QA is seen.

**Scaling with encoder size.** After investigating the impact of the LLM decoder in Fig. 2, we examine the impact of increasing the vision encoder size from 300M (PE Large) to 2B (PE Giant) for each language model scale next. In Fig. 9, we overlay the new power-law with the 2B vision encoder (**black** dashed) line onto the 300M (**red** dashed) line. Notably, we find that the larger vision encoder (300M → 2B) leads to greater scaling trend on video QA benchmarks. Quantitatively, the power law fit has improved from $-0.15$ to $-0.19$. The two lines intersect around 8B scale with PE-G, proving that 8B and larger PLM will benefit more with larger vision encoder. We use PE-L for 1B and 3B LLM scale and PE-G for 8B scale by default.

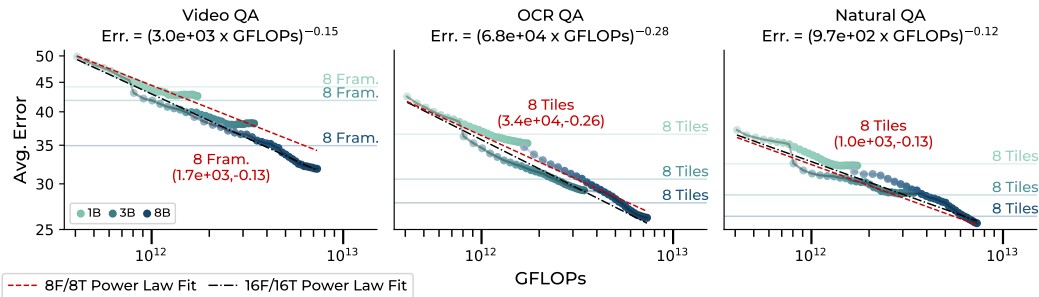

Figure 10: **Scaling with input size.** Scaling trends of training with 16 tiles/frames vs. 8 tiles/frames. Higher input size scales better in Video QA and OCR QA tasks while similar trend is seen for Natural QA.

**Scaling with input size.** In Fig. 10, we show the impact of increasing the input size to VLM through higher image resolution and more video frames. In this setting, each scale of PLM trains with *dynamic tiling* for image input and *uniform sampling* for video input with maximum 8 or 16 tiles/frames per sample. In each plot, the average error of PLM trained with 16 tiles/frames are plotted. All models use $2 \times 2$ spatial average pooling before input to LLM, and each tile/frame has $448 \times 448$ resolution. Similar to Fig. 2, we show power law fit with a **black** dashed line, and compare to 8 tiles/frames training denoted with **red** dashed line. Notably, we find out that on Video QA and OCR QA benchmarks, PLM shows better scalability with training with higher input size. This means *with the same FLOP counts at $10^{13}$, training with 16 frames makes 2.0 points of metric error lower than 8 frames counterpart* (32.2 vs 30.2). Similar trends are observed with OCR QA going from 8 tiles max. to 16 tiles max. Notably, higher resolution did not make a difference for Natural QA tasks. We chose the 16 max-tiles and frames to be our final training setting for stage 2 PLM.

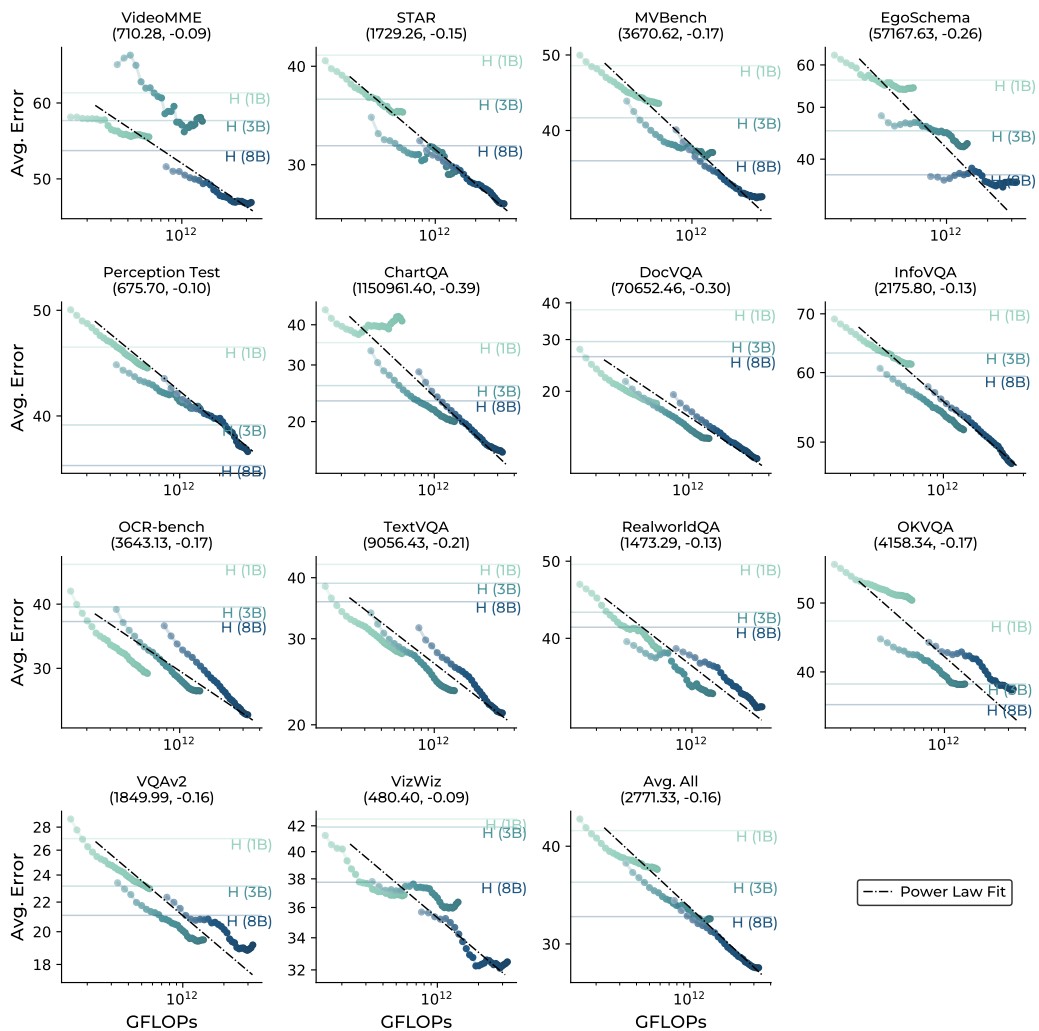

Figure 11: **Synthetic Scaling Plots.** Relationship between Average Error and training compute (in floating-point operations) for various 1B, 3B, 8B PLM with L14 vision encoder. Each plot reports the individual error in VideoMME [20], STAR [21], EgoSchema [22], How2QA [8], MVBench [23], PerceptionTest [24], ChartQA [25], DocVQA [26], InfoVQA [27], OCRBench [28], RealworldQA [29], OKVQA [30], VQAv2 [31], VizWiz [33], and TextVQA [34]. Finally, we report Avg. All, which average over all the metrics.

In Fig. 11, we show the breakdown of the scaling trend shown in §3.2. "H" stands for *human only* (*i.e.*, no synthetic) baseline. From the breakdown, the most notable point is the the scalability in OCR, Chart, Document QA tasks. In each benchmark, synthetic data makes more than 10 points of improvement on every model scale, compared to "no synthetic" baselines. Moreover, there is no sign of saturation; the performance will most likely improve with more synthetic data. We hypothesize that OCR, Chart, Document QA tasks reduce to "translation" task — a set of pixels has one-to-one mapping to text space. Remaining tasks exhibit clean power-law relationship between metric error and FLOPs. The last plot shows scaling trend on average over all benchmarks, which shows a close power-law relationship.

# C VLM Benchmark Details

In this section, we provide details about all the image and video benchmarks considered in §5 including composition and evaluation metrics for image benchmarks (§C.1), video benchmarks (§C.2) and our PLM–VideoBench (§C.3. We also describe evaluation protocol for all these benchmarks including inference parameters and prompts (§C.4). Pointers to evaluation code are linked where available.

## C.1 Image Benchmarks

**Image captioning** We evaluate on single image captioning and grounded image captioning benchmarks like COCO [101], nocaps [102] and Flickr [103]. We report CIDEr as the evaluation metric.

**Perception and reasoning** We evaluate on broad, general purpose VQA benchmarks like MMMU [68], VQAv2 [31], MMBench [94], OK-VQA [69], VizWiz [33] as well as hard perception benchmarks like BLINK [71], CV-Bench [72], RealWorldQA [29], and VSR [73]. For all MCQ benchmarks, we report accuracy of selecting the correct option.

**Charts, diagrams and documents** We evaluate on benchmarks for reasoning over various types of charts, graphs, diagrams, infographics etc. Specifically, DocVQA [26], ChartQA [65], TextVQA [66], InfographicsVQA [27], AI2D [67], OCRBench [28], and SEED [70]. We report accuracy of selecting the correct option.

**Image Hallucination** Finally, we evaluate on benchmarks that evaluate robustness of models to hallucinated details in questions such as HallusionBench [111] and POPE [32]. For HallusionBench we report the *aAcc* metric (code) which accounts for correctness and consistency using an LLM judge.

## C.2 Video Benchmarks

**Video captioning** We evaluate on short-video captioning benchmarks, namely YouCook2 [47] and VATEX [59] as well as recent detailed video captioning benchmarks — DREAM-1k [74] and AuroraCap-VDC [121]. For YouCook2 and VATEX, we report CIDEr score [208]. For DREAM-1k we report AutoDQ F1-score (code) and for AuroraCap-VDC we report the VDC accuracy (code) following the author's proposed metric.

**Short video QA** We evaluate on multiple-choice (MCQ) benchmarks such as How2QA [8], NExt-QA [50], PerceptionTest [24], STAR [21], TGIF-QA [112], TVQA [113], Video-MME [20] and TVBench [117]. We report accuracy of selecting the correct option. We also evaluate on open-ended question answering benchmarks (w/o options) such as ActivityNet-QA [51] (code), MMBench-Video [116] (code) and VCGBench-Diverse [84]. We report LLM-judge scores/accuracies for these benchmarks. For VCGBench-Diverse, we report the average of 5 LLM-judge scores (code).

**Long video QA** We evaluate on popular long-video benchmarks such as EgoSchema [22], LVBench [41], LongVideoBench [123] and MLVU [125]. We report accuracy of selecting the correct option.

**Fine-grained video QA** We evaluate on benchmarks for fine-grained spatial, temporal and detail reasoning in videos such as TemporalBench [39], TOMATO [36], MotionBench [38], TempCompass [75] and CG-Bench [35]. We report accuracy of selecting the correct option. For TemporalBench, we report the *multi-binary accuracy* (MBAcc) (code) proposed by the authors to reduce bias in evaluation.

**Hallucination** We evaluate on benchmarks that evaluate robustness of models to hallucinated details in questions such as VideoHallucer [76] and EventHallusion [37]. We report accuracy of selecting the correct option.

## C.3 PLM-VideoBench

We evaluate on our suite of benchmarks for fine-grained and spatio-temporal reasoning in videos. These include:

**Fine-grained QA (FGQA)** We report multi-binary accuracy (MBAcc) following prior work [39]. In short, this entails presenting the model multiple independent, binary-choice questions about the same video (in our case, three questions) and requiring the model to gets all of them correct, to count towards accuracy. This sets a higher bar for models, and combats bias in multiple-choice question benchmarks that prior work identifies.

**SmartGlasses-QA (SGQA)** We report LLM-judge accuracy of the predicted answer compared to the ground truth answer. We follow existing LLM judge prompts from ActivityNetQA (code). The prompt is repeated below for completeness.

**Video Region Captioning (PLM-RCap)** We use an LLM-judge to generate the similarity scores between predicted and ground truth captions. The prompt is below.

**Dense Video Region Captioning (PLM-RDCap)** We adapt the SODA metric [60] from dense video captioning literature for this task. To compute this metric, we use the same LLM-judge from above to generate the pairwise similiarity scores between predicted and ground truth captions, which is then fed to the standard metric computation routine.

**Region Temporal Localization (PLM-RTLoc)** We report standard temporal localization metrics, namely Mean Recall@1, averaged over a range of IoU thresholds $[0.3, 0.5, 0.7, 0.9]$.

## C.4 Evaluation Protocols

**Common evaluation protocol.** For video benchmark evaluations, we sample 32 frames uniformly from the full video unless otherwise specified. For uniformity and consistency across benchmarks, we implement all LLM-judge evaluations using LLama3.3-70B-Instruct [13], following LLM judge prompts from popular evaluation frameworks [209, 210] where available. Outputs from all models are generated via greedy sampling (temperature 0).

```
SG-QA judge prompt

You are an intelligent chatbot designed for evaluating the correctness of generative
outputs for question-answer pairs. Your task is to compare the predicted answer with
the correct answer and determine if they match meaningfully.  Here's how you can
accomplish the task:
––
##INSTRUCTIONS:
- Focus on the meaningful match between the predicted answer and the correct answer.
- Consider synonyms or paraphrases as valid matches.
- Evaluate the correctness of the prediction compared to the answer.

Please evaluate the following video-based question-answer pair:
Question: [question]
Correct Answer: [target]
Predicted Answer: [candidate]
Provide your evaluation only as a yes/no and score where the score is an integer value
between 0 and 5, with 5 indicating the highest meaningful match. Please generate the
response in the form of a Python dictionary string with keys 'pred' and 'score', where
value of 'pred' is a string of 'yes' or 'no' and value of 'score' is in INTEGER, not
STRING. DO NOT PROVIDE ANY OTHER OUTPUT TEXT OR EXPLANATION. Only provide the Python
dictionary string. For example, your response should look like this: {"pred": "yes",
"score": 4.8}.
```

```
PLM-RCap judge prompt

Your task is to compare a given pair of captions and provide a single score indicating
how correct the pred is compared to GT, on a scale from 0 to 10.  Focus on meaning
and context, not exact word matches. Penalize missing and incorrect information, with
lower scores for more significant errors. High scores require accurate conveyance of
all key GT information.  Respond with only the score, starting your response with the
number and including no additional text. Output format: [score].
```

**PLM-VideoBench inference prompts.** Table 11 contains example inference prompt examples for each PLM-VideoBench task. Note that some variation exists between instances in the benchmark. For

example, for RCap a prompt may be "What is happening to the subject in the region highlighted by the red rectangle ..." instead of "Give a detailed description of the events occurring in the region marked by the red rectangle ...", however they convey the same underlying instruction and information.

Proprietary models like GPT-4o and Gemini require more careful prompting to ensure that the output formatting is respected. For example, we append instructions to prevent model hallucinations (*e.g.*, "You must use these frames to answer the question; do not rely on any external knowledge or commonsense."), to prevent refusals to answer (*e.g.*, "Even if the information in these separate frames is not enough to answer the question, please try your best to guess an answer which you think would be the most possible one based on the question. Do not generate answer such as *not possible to determine*") and in-context examples to help guide the model towards the correct output format. Model- and benchmark-specific inference prompts will be released along with our code for full reproducibility.

| Task | Prompt |
|------|--------|
| FGQA | Question: [question] \n Options: \n (A) [option1] \n (B) [option2] \n Only give the best option. |
| SGQA | The following question is asked by the camera wearer at the end of the video. Provide a detailed answer even if unsure. Try to answer in around 20-30 words. Now answer the following question based on the video content: [question] |
| RDCap | Create a dense caption of the subject's actions within the red rectangles, including action frames ids and brief descriptions. For each item use the format [start, end]: [description] separated by a newline, where start and end are frame numbers between 0 and 31 in this 32 frame video. |
| RCap | Give a detailed description of the events occurring in the region marked by the red rectangle within frames ([start frame], [end frame]) in this 32 frame video |
| RTLoc | Given the region marked by the red rectangle in the video, please provide the start and end frame of when '[event]' happens. Use the format (start, end), where start and end are frame numbers between 0 and 31 in this 32 frame video. |

Table 11: **PLM-VideoBench task prompts.** Items in square brackets are placeholders filled in for each benchmark instance.

# D    Baseline Implementation Details

We provide baseline-specific implementation details for all models in §5.1 of the main paper.

**Proprietary baselines** We evaluate the GPT and Gemini family of models. For GPT-4o, we use the `GPT-4o-2024-11-20` checkpoint . We feed 32 uniformly sampled frames regardless of video length, loaded at *high* image quality setting. For Gemini, we evaluate Gemini-1.5-Pro and Gemini-2.0-Flash. For VQA tasks, we input the video (without audio) which is processed internally at 1 fps. For spatio-temporal tasks (RCap, RDCap, and RTLoc) we use the same inputs as for open-source models and GPT-4o. We evaluate these models using API call.

**Open-source models** We evaluate InternVL, Qwen, Molmo and Llava-OV models. We follow official implementation and preprocessing pipelines for each. Specifically, we evaluate InternVL2 and InternVL2.5 (code); QwenVL2 and QwenVL2.5 (code); Molmo-O-0924 (code) and Llava-OV (code). For QwenVL, we sample frames at 1 fps from videos. For InternVL2, we use 12 tiles per image as this more closely matches the reported results.

**Human performance baseline.** In Table 5, we report human performance on PLM-VideoBench. For each task, we present annotators with the test sets and collect answers for each instance given the standard task prompt. Given the difficulty of RDCap, we reuse our data annotation pipeline in §G to collect new dense captions independently, rather than providing the standard task instruction.

# E  Additional Results

## E.1  Additional PLM-VideoBench Results

We present benchmarking results across all model scales (1B, 3B, 8B) in Table 12, to supplement the 8B model results in the main paper (Table 5). Our approach consistently outperforms baselines across all scales, including proprietary models whose model scale is unknown.

| Model | FGQA *MBAcc* | SGQA *acc†* | RDCap *SODA†* | RCap *score†* | RTLoc *meanR* | Avg. |
|---|---|---|---|---|---|---|
| Human perf. | 90.9 | 67.9 | 66.6 | 53.9 | 67.8 | 70.9 |
| **Proprietary** | | | | | | |
| GPT-4o [61] | 61.2 | 63.7 | 20.9 | 35.7 | 33.1 | 51.6 |
| Gemini 1.5 Pro [63] | 57.1 | 49.9 | 14.4 | 33.1 | 27.6 | 44.0 |
| Gemini 2.0 Flash [63] | 58.7 | 44.8 | 13.2 | 30.9 | 27.6 | 42.5 |
| **1B scale** | | | | | | |
| Qwen2VL-2B [16] | 39.0 | 38.5 | 0.9 | 18.1 | 10.8 | 29.1 |
| InternVL2-1B [10] | 35.8 | 28.9 | 0.3 | 17.2 | 2.7 | 23.8 |
| InternVL2.5-1B [10] | 42.3 | 39.6 | 6.7 | 23.6 | 1.6 | 30.8 |
| PLM-1B | **57.6** | **40.9** | **50.3** | **40.9** | **57.7** | **49.4** |
| **3B scale** | | | | | | |
| Qwen2.5 VL-3B [15] | 43.7 | 45.1 | 0.3 | 17.2 | 13.9 | 33.1 |
| InternVL2-4B [10] | 43.2 | 41.7 | 0.5 | 19.9 | 9.6 | 30.3 |
| InternVL2.5-4B [10] | 50.0 | **49.2** | 4.9 | 25.9 | 15.4 | 35.3 |
| PLM-3B | **67.1** | 38.8 | **53.1** | **45.0** | **58.2** | **53.0** |
| **8B scale** | | | | | | |
| LLaVA-OV-7B [64] | 40.2 | 41.5 | 4.7 | 24.4 | 13.9 | 32.0 |
| Qwen2VL-7B [16] | 49.2 | 44.5 | 4.1 | 17.6 | 15.1 | 35.3 |
| Qwen2.5VL-7B [15] | 49.8 | 43.0 | 2.5 | 21.5 | 10.7 | 34.8 |
| InternVL2-8B [10] | 47.7 | 45.9 | 1.2 | 21.5 | 11.6 | 35.0 |
| InternVL2.5-8B [10] | 53.7 | **48.3** | 5.7 | 26.1 | 8.8 | 38.5 |
| PLM-8B | **67.7** | 46.2 | **52.8** | **46.6** | **59.1** | **55.6** |

Table 12: **PLM-VideoBench results** across all model scales to supplement results in Table 5.

## E.2  Comparison with LLaMA-3V

| Model | Avg. | DocVQA *(test) acc* [26] | ChartQA *acc* [65] | TextVQA *acc* [66] | InfoQA *(test) acc* [27] | AI2D *(w/o mask) acc* [67] | MMMU *(val) acc* [68] | VQAv2 *(val) acc* [31] |
|---|---|---|---|---|---|---|---|---|
| LLaMA 3.2V (11B) [13] | 73.0 | 88.4 | 83.4 | 79.7 | 63.6 | 91.1 | 50.7 | 75.2 |
| LLaMA 3.2V (90B) [13] | 76.6 | 90.1 | 85.5 | 82.3 | 67.2 | 92.3 | 60.3 | 78.1 |
| PLM-1B | 67.1 | 90.7 | 78.6 | 82.1 | 63.0 | 84.9 | 34.8 | 81.7 |
| PLM-3B | 74.4 | 93.8 | 84.3 | 84.3 | 74.6 | 90.9 | 41.2 | 84.3 |
| PLM-8B | 76.2 | 94.6 | 86.5 | 86.5 | 80.9 | 92.7 | 46.1 | 85.6 |

Table 13: **PLM versus LLaMA-3V on Image Benchmarks:** Note that we use LLaMA-3V-90B [13] for generating image captions in our synthetic data engine.

## E.3  Image Captioning

Table 14 shows PLM versus proprietary models and open-access baselines of comparable scale on Image Captioning benchmarks.

## E.4  Image Grounding

Table 15 show PLM versus SoTA model on RefCOCO/+/g. PLM performs competitively compared to the baselines across all model scales, and outperforms specialist models.

## E.5  Long Video Understanding

Table 16 show PLM versus open-access baselines and proprietary models of comparable scale, and report results over 3 long video QA benchmarks.

| Model | COCO (karpathy) CIDEr [101] | Nocap CIDEr [102] | Flickr CIDEr [103] |
|---|---|---|---|
| **Proprietary** | | | |
| GPT-4o [61] | 74.4 | 76.6 | 71.7 |
| Gemini 1.5 Pro [63] | 70.6 | 71.1 | 68.2 |
| Gemini 2.0 Flash [63] | 84.8 | 85.0 | 66.6 |
| **1B scale** | | | |
| Qwen2VL-2B [16] | 107.1 | 101.2 | 86.0 |
| InternVL2.5-1B [10] | 122.6 | 110.5 | 86.1 |
| PLM-1B | **138.6** | **124.2** | **100.5** |
| **3B scale** | | | |
| Qwen2.5 VL-3B [15] | 101.7 | 105.5 | 77.5 |
| InternVL2.5-4B [10] | 125.4 | 117.1 | 87.4 |
| PLM-3B | **144.9** | **126.5** | **98.0** |
| **8B scale** | | | |
| LLaVA-OV-7B [64] | 112.1 | 70.7 | 55.7 |
| Qwen2.5VL-7B [15] | 36.8 | 32.7 | 34.9 |
| InternVL2.5-8B [10] | 125.8 | 116.7 | 96.5 |
| PLM-8B | **146.7** | **129.9** | **105.6** |

Table 14: **Image Captioning benchmarks.** PLM versus proprietary models and open-access baselines of comparable scale on Image Captioning benchmarks.

| Model | RefCOCO val | RefCOCO testA | RefCOCO testB | RefCOCO+ val | RefCOCO+ testA | RefCOCO+ testB | RefCOCOg val | RefCOCOg test | Avg. |
|---|---|---|---|---|---|---|---|---|---|
| **Specialists** | | | | | | | | | |
| GroundingDINO [211] | 90.6 | 93.2 | 88.2 | 88.2 | 89.0 | 75.9 | 86.1 | 87.0 | 86.6 |
| UNINEXT-H [212] | 92.6 | 94.3 | 91.5 | 85.2 | 89.6 | 79.8 | 88.7 | 89.4 | 88.9 |
| ONE-PEACE [213] | 90.6 | 93.2 | 88.2 | 88.2 | 89.0 | 75.9 | 86.1 | 87.0 | 86.6 |
| **1B scale** | | | | | | | | | |
| PLM-1B | 88.5 | 91.5 | 84.8 | 83.2 | 88.6 | 76.5 | 86.0 | 86.4 | 85.7 |
| **3B scale** | | | | | | | | | |
| Qwen2.5 VL-3B [15] | 89.1 | 91.7 | 84.0 | 82.4 | 88.0 | 74.1 | 85.2 | 85.7 | 85.0 |
| PLM-3B | **93.3** | **94.9** | **89.5** | **89.8** | **93.6** | **84.2** | **90.8** | **90.9** | **90.9** |
| **8B scale** | | | | | | | | | |
| Cube-LLM [214] | 90.9 | 92.6 | **87.9** | 83.9 | 89.2 | 77.4 | 86.6 | 87.2 | 87.0 |
| Qwen2VL-7B [16] | **91.7** | 93.6 | 87.3 | 85.8 | 90.5 | 79.5 | 87.3 | 87.8 | 87.9 |
| Qwen2.5VL-7B [15] | 89.1 | 91.7 | 84.0 | 82.4 | 88.0 | 74.1 | 85.2 | 85.7 | 85.0 |
| InternVL2-8B [10] | 87.1 | 91.1 | 80.7 | 79.8 | 87.9 | 71.4 | 82.7 | 82.7 | 82.9 |
| InternVL2.5-8B [10] | 90.3 | **94.5** | 85.9 | 85.2 | **91.5** | 78.8 | 86.7 | 87.6 | 87.6 |
| PLM-8B | 90.6 | 91.8 | 85.9 | **87.3** | 91.3 | **81.1** | **88.8** | **89.2** | **88.2** |

Table 15: **Image Grounding results on RefCOCO/+/g.** PLM performs competitively compared to the baselines across all model scales, and outperforms specialist models for the image grounding task.

## E.6 Additional Ablations

Table 17 shows the performance of PLM under full stage-3 setup, and Table 18 shows the effect of removing human-annotated in-distribution data in PLM-VideoBench.

| Model | Long Video QA | | |
|---|---|---|---|
| | LVBench *acc* [41] | LongVideoBench *(val) acc* [123] | MLVU *(dev) M-avg* [125] |
| **Proprietary** | | | |
| GPT-4o [61] | 37.2 | 66.7* | 67.4 |
| Gemini 1.5 Pro [63] | 33.1* | 64.0* | 69.9 |
| Gemini 2.0 Flash [63] | - | 61.6* | 69.5 |
| **1B scale** | | | |
| Qwen2VL-2B [16] | **42.0** | 47.9 | **62.7** |
| InternVL2-1B [10] | 31.4 | 43.3* | 52.0 |
| InternVL2.5-1B [10] | 35.3 | 47.9 | 57.3* |
| PLM-1B | 40.0 | **52.3** | 58.9 |
| **3B scale** | | | |
| Qwen2.5 VL-3B [15] | **43.3*** | 54.2* | 68.2 |
| InternVL2-4B [10] | 34.0 | 53.0* | 59.9* |
| InternVL2.5-4B [10] | 40.1 | 56.3 | **68.3*** |
| PLM-3B | 40.4 | **57.9** | 65.0 |
| **8B scale** | | | |
| LLaVA-OV-7B [64] | 38.8 | 55.7 | 64.6 |
| Qwen2VL-7B [16] | **46.0** | 55.8 | 69.8* |
| Qwen2.5VL-7B [15] | 45.3* | 56.0* | **70.2*** |
| InternVL2-8B [10] | 37.0 | 55.4 | 64.0* |
| InternVL2.5-8B [10] | 43.2* | **60.0*** | 68.9 |
| PLM-8B | 44.5 | 56.9 | 66.4 |

Table 16: **Results on long video understanding tasks.** We compare PLM with open-access baselines and proprietary models of comparable scale, and report results over 3 long video QA benchmarks. Cells with * are reported numbers from literature. The remaining are reproduced using official code.

| Model | Avg. | DocVQA *(test) acc* [26] | ChartQA *acc* [65] | TextVQA *acc* [66] | InfoVQA *(test) acc* [27] | AI2D *(w/o mask) acc* [215] | OCRBench *acc* [28] | MMMU *(val) acc* [68] | VQAv2 *(val) acc* [31] | OK-VQA *acc* [30] | VizWiz *acc* [33] | SEED *(image) acc* [70] | BLINK *(multi-image) acc* [71] | CV-Bench *acc* [72] | RealWorldQA *acc* [29] | VSR *acc* [73] | POPE *acc* [32] |
|---|---|---|---|---|---|---|---|---|---|---|---|---|---|---|---|---|---|
| Stage2+3 | 75.3 | 91.8 | 82.6 | 84.5 | 71.2 | 90.0 | 83.7 | 39.9 | 83.6 | 67.4 | 62.8 | 78.5 | 54.6 | 72.5 | 72.8 | 79.9 | 89.6 |
| PLM-3B | 76.5 | 93.8 | 84.3 | 84.3 | 74.6 | 90.9 | 83.0 | 41.2 | 84.3 | 66.8 | 64.0 | 78.5 | 55.4 | 81.4 | 72.4 | 80.4 | 88.7 |

| Model | Avg. | MVBench *acc* [23] | NExT-QA *acc* [50] | Percept.Test *(test) acc* [24] | STAR *acc* [21] | VideoMME *acc* [20] | ActNetQA *acc* [51] | EgoSchema *(test) acc* [22] | TempBench *MBAcc* [39] | TOMATO *acc* [36] | MotionBench *(dev) acc* [38] | TempCompass *(MCQ) acc* [75] | CG-Bench *(clue) acc* [?] | Charades-STA *mIOU* [40] | VideoHallucer *overall acc* [76] | EventHallusion *(binary) acc* [37] |
|---|---|---|---|---|---|---|---|---|---|---|---|---|---|---|---|---|
| Stage2+3 | 60.5 | 71.6 | 84.4 | 73.9 | 82.9 | 56.7 | 67.5 | 71.8 | 25.7 | 28.5 | 57.2 | 69.8 | 44.9 | 47.0 | 53.4 | 71.9 |
| PLM-3B | 62.5 | 74.7 | 83.4 | 79.3 | 84.8 | 54.9 | 66.2 | 72.6 | 23.4 | 30.9 | 60.4 | 69.3 | 47.2 | 57.7 | 55.5 | 76.5 |

Table 17: **Performance comparison under full stage-3 setup.** We run PLM-3B using a combined 90M+ data blend with up to 36 image tiles and 32 video frames (i.e., stage-3 configuration) and compare against other open multimodal models. PLM-3B achieves the best overall performance across reasoning, captioning, and localization benchmarks, highlighting its superior generalization and cross-modal understanding.

| Model | FGQA *MBAcc* | SGQA *acc†* | RDCap *SODA†* | RCap *score†* | RTLoc *meanR* | Avg. |
|---|---|---|---|---|---|---|
| Qwen2.5-VL (3B) | 43.7 | 45.1 | 0.3 | 17.2 | 13.9 | 24.0 |
| InternVL2.5 (4B) | 50.0 | 49.2 | 4.9 | 25.9 | 15.4 | 29.1 |
| PLM (3B, no hum.) | 54.0 | 41.5 | 11.1 | 27.6 | 25.0 | 31.8 |
| PLM (3B) | 67.1 | 38.8 | 53.1 | 45.0 | 58.2 | 52.4 |

Table 18: **Effect of removing human-annotated in-distribution data in PLM-VideoBench.** We retrain PLM-3B after excluding all human-annotated samples from in-distribution datasets. Despite this stricter setting, PLM outperforms Qwen2.5-VL (3B) and InternVL2.5 (4B) on most metrics and achieves the best overall average, indicating stronger generalization without in-domain supervision.

## F  PLM–FGQA: Fine-grained QA

We present PLM–FGQA Fine-grained QA (FGQA), a video dataset focused on "how" actions are performed, capturing nuanced fine-grained details through specially designed questions and carefully annotated answers. Due to the scarcity of fine-grained video Q&A data, see Table 19, we built a data engine to enable the collection of our 2.4M Q&A dataset, PLM–FGQA. As illustrated by the breakdown of question types[3] in Fig. 4 (top-right), PLM–FGQA contains a large number of annotations about fine-grained details that have been largely missing in existing training video QA datasets [24, 49–56].

| Dataset | Year | #Q&As | Dataset | Year | #Q&As |
|---|---|---|---|---|---|
| MovieQA | 2016 | 6462 | STAR | 2021 | 60000 |
| MSRVTT-QA | 2017 | 243690 | CLEVRER | 2023 | 82620 |
| TGIF-QA | 2017 | 165165 | EgoQA | 2024 | 19000 |
| MSVD-QA | 2017 | 51000 | PerceptionTest | 2024 | 44146 |
| TVQA | 2018 | 152545 | VideoInstruct | 2024 | 25803 |
| ActivityNetQA | 2019 | 58000 | MoVQA | 2024 | 21953 |
| How2QA | 2020 | 44007 | CinePile | 2024 | 303828 |
| NexT-QA | 2021 | 52044 | Sports-QA | 2025 | 94000 |
| PLM-FGQA | 2025 | 2379067 | | | |

Table 19: Comparison of our PLM–FGQA dataset with existing video-QA datasets.

### F.1  Annotation process: Data Engine

Our data engine is built upon the following modules: (1) Temporal Segment Generation, (2) Question Generation, (3) Answer Generation, (4) Human Annotation (answer verification/manual answer annotation), (5) Quality Control, as illustrated in Figure 12. Next, we describe each module in detail, and finally also provide additional details about the extra steps we took for forming the FG-QA component of PLM–VideoBench out of these annotations.

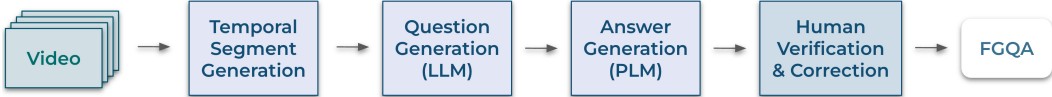

Figure 12: Data engine used to collect the PLM–FGQA dataset.

### F.1.1  Temporal Segment Generation

We source the video data that serves as a basis for our annotations from publicly available datasets. Based on the video sources and the type of existing annotations, we split the videos into three distinct categories.

**Videos with existing ground-truth segment annotations**: We directly adopt segments with their human-annotated action annotations from the following datasets: Ego4d Goal-Step[216], Ego4D Moments[43], EgoExo4D [44], HT-Step[217, 218], COIN [45], CrossTask [46], and YouCook2 [47]. All those sources provide video segment boundaries accompanied by some form of textual action descriptions, and are therefore readily usable with the rest of the pipeline.

**Unedited videos of physical activities**: For physical activities videos (*e.g.* basketball, dancing, soccer), actions are usually atomic and short (*e.g.* dribble, dance move, kick) and therfore reuqire precise temporal localization. To source videos for these scenarios we used data from EgoExo4D [44] that contains temporally well-aligned and precise narrations; we obtained segments of 2-3 seconds centered around narration timings, and used the anchor narrations directly as the action description.

**Raw, untrimmed videos in-the-wild without temporal segment annotations**. We source a very large part of our data from untrimmed instructional videos in the large-scale HT100M dataset [42] which we first need to segment before use. The goal is to obtain video clips that contain meaningful,

---

[3]obtained with LLM-based tagging.

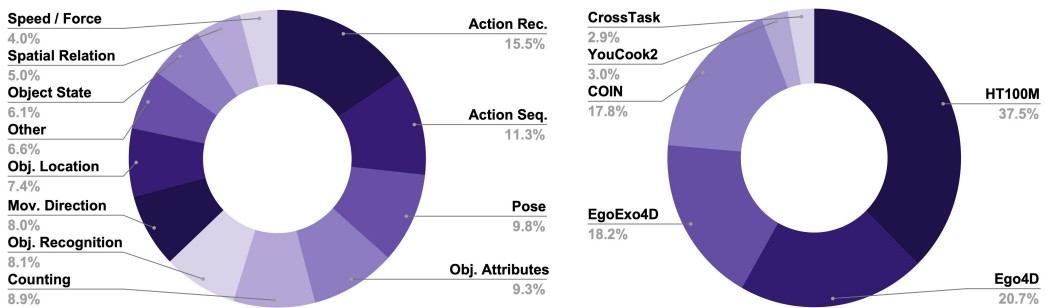

Figure 13: Distribution of question types (left) and video sources (right) in the FGQA component of PLM–VideoBench.

salient actions, and also caption the resulting segments with concise but accurate action descriptions. We describe the automatic segmentation and captioning module in the following.

The automatic segmentation and captioning pipeline involves the following three stages:

**Temporal segment proposal.** Given untrimmed long videos, the first step is to identify semantically coherent segments within them. Inspired by prior work on unsupervised action proposal and segmentation, we leverage visual feature clustering to generate temporal segment proposals, and use shot-boundary detection results to further refine the segment boundaries. We extract clip-level visual features[219] using a sliding window with temporal stride of 1 second. We then compute the pairwise similarity between neighborhood features and detect the class-agnostic action boundaries using a boundary detection kernel (similar to those used in literature[220, 221]). Finally, since the detected segments are usually over-segmented, we perform a bottom-up agglomerate clustering approach to group adjacent segments into clusters, using a segment duration prior of 10 seconds. We also leverage shot boundary detection[222] to obtain precise moments of scene changes: we refine the boundaries of the segment proposals by aligning them to the detected shot boundaries when they're sufficiently close ($\leq 1$ second).

**Segment filtering and ranking.** How-to videos often include a lot of content that is irrelevant to the demonstration of the activity at hand, such as the instructor explaining what they are about to do or showcasing tools and ingredients. It is therefore important to detect and filter segments with such uninformative content. To that end we rank candidate segments according to relevance using a series of heuristics and learned models, described bellow.

*a. Talking head detection.* A common mode in instructional videos is instructors talking into the camera, describing objects or explaining actions they're about to take. To detect and remove such segments, we employ an Active Speaker Detection (ASD) pipeline[223], which we run densely on every video and combine resulting talking head tracks, to produce an ASD score for every segment.

*b. Hand-object interaction (HOI) detection.* The presence of hand-object interaction (HOI) can be a good indicator of visually groundable actions. We leverage the temporal selection strategy[224] to filter out the segment proposals that contain hand-object interaction. We first employ an off-the-shelf robust HOI detector[225] to densely extract HOI regions within a proposed segment. The HOI score is then calculated by measuring the likelihood of hand-object interaction in the segment and the averaged probability of all the detected hands.

*c. ASR groundability.* HT100M videos contain timestamped ASR captions, which are speech transcriptions of the audio instructions. It is desirable to rank candidate segments based on how likely their ASR content is to their video content. The hypothesis here is that segments containing ASR transcriptions that align well to the video content, are more likely to be visual-information rich. Moreover since the action labeling pipeline (described next) relies on ASR metadata for producing descriptions, higher ASR groundability scores make it likelier to produce good quality segment descriptions. For every candidate segment, we compute an ASR-groundability score by computing video-text alignment scores[219] for each ASR caption within the segment and then averaging the ones that are above a threshold (we use 0.5).

*d. Relevance classification.* The above heuristics work well for the clear-cut cases they are tailored for, but in practice we found that they struggle with more nuanced segments (*e.g.* instructor fiddling with an object and describing it rather than using it). To improve the detection of those cases, we manually labelled a small amount of segments that passed through the other filters and trained a binary classifier to classify them as "relevant" or "irrelevant"; to that end we trained a simple 2-layer MLP classifier on top of temporally pooled video representations with a logistic loss for binary classification. We deployed the trained model to provide a relevance score for all the candidate segments.

We combined the scores resulting from all the modules described above and determined cutoff thresholds, based on a small manually annotated validation set. In production, we keep all the segments that have relevance scores above those thresholds.

**Segment captioning** We follow a two-step process to obtain action labels for each unlabeled segment: In the first step, a collection of off-the-shelf perception models are used to extract individual image-level captions, video-level captions, and object detections from the segment. The output of all perception models is then fed as text into an LLM to generate long, fine-grained captions. At the second step, the detailed captions are fused with the ASR content of the segment, to obtain a consice action description. Specifically, we query an LLM (Llama 3.3 70B [13]) with the following prompt:

```
Segment to action labels prompt

Detailed description: [fine grained caption] ASR transcription: [asr caption]. Given
the detailed description above, identify the specific action performed as part of the
activity [task name]. Your response must not be the same as the activity [task name]
and needs to be a specific substep within the activity [task name]. Please also supply
a rationale for your answer.
```

The extracted labeled video segments obtained through the above process serve as the foundation for the subsequent Q&A generation.

### F.1.2 Automatic Question Generation

We automatically generate questions about the fine-grained details of the way activities are executed in the video. Our questions is generated with a variety of prompts and models which lead to increased question diversity and specificity. In Table 20 we present the question types and sample questions per question type. Here, we summarize how these questions are generated automatically with an ensemble with models and prompts:

**LLM-based action-conditioned question generation** Given a segment, its action name (*e.g.*, *cut potatoes*), a task name (*e.g.*, *How to make sweet potato gratin*) and optionally other metadata about the segment (for example, recognized objects [**?** ]), we generate questions that can elicit descriptions of fine-grained details by raters with an LLM. We use tailored prompts for generating questions that cover *how* the activity is executed (tools, object locations, object states, direction of movements, hand pose), and the spatial arrangement of objects.

```
Activity FG question generation prompt

I am learning how to [action name] while [task name]. Ask me [N] most relevant questions
that reveal the details of the way the step is executed in my environment, e.g., (a)
part location, (b) types of tools/ingredients used, (c) direction of movements, (d)
how are objects held, (e) object states at the beginning of the step, (f) object state
at the end of the step.  The questions must be answerable by visually observing the
activity, without reading instructions or trying out.  Please indicate the type of
question from (a) to (f) for each question asked at the beginning of the question.
```

```
Spatial FG question generation prompt

Imagine I have no common sense or understanding of the 3D real world. I am trying to
[task name] and am at the step where I am [action name]. There's [object list] when I'm
[action name]. Ask me [N] questions about the 3D position of objects, relative location
between objects, distance between objects, spatial relationship using prepositions like
above, below, next to, etc. that I might want to know. The questions must be answerable
by only visually observing me performing activity, without reading instructions or
trying out.
```

We explicitly encourage the LLM to provide questions that can be answered solely based on the video frames, in contrast to questions that are focused on external knowledge or non-groundable concepts or judging the execution of the step (*e.g.*, avoid questions like *is the pan hot enough to add the oil?*, *what tool is typically used to loosen the axle nut*). The rationale for this is to collect as many Q&A pairs that a model cannot answer just based on external knowledge/language prior, but they rather require vision perception to be answered. Note that these questions are generated without visual input, hence they are not instance-specific and might not be answerable given the video segment.

**VLM-based instance-specific question generation** After collecting a first set of Q&As using the LLM-generated questions, we bootstrap a VLM Question Generator model, which takes as input the video segment, question types and optionally the task name, and generates a set of instance-specific visual questions. The VLM Question Generator model is obtained by supervised fine-tuning of PLM with a question generation instruction-tuning dataset which consists of triplets (video, prompt, response), where the prompt includes the instruction to generate questions based on question types and the response includes example questions to be generated for the given video. Due to the lack of such a dataset with fine-grained question, we synthetically generated it by utilizing the Q&A pairs obtained based on the LLM-generated questions. Specifically, for each video segment, we use an LLM to (1) decompose existing Q&A pairs into multiple Q&A pairs, with each new question focusing on one detail of the original answer; (2) tag question types for the generated questions based on an expanded list of question types; and (3) generate a (prompt, response) pair for the segment. This resulted in $\sim 600k$ training instances.

```
VLM Question Generator training sample

Generate 3 different questions that reveal the fine-grained details of the way the
activity is executed. In particular, focus on these question types: fine-grained object
locations, hand pose, object/repetition counts, generating at least one question per
type. Write each question in a separate line, e.g., Q1. first question.
Q2. second question.
...
QN. N-th question.
Response:
Q1. Where are the tomatoes positioned prior to being cut?
Q2. How is the person grasping the tomato with their left hand?
Q3. How many tomatoes did the person use in the segment?
```

**LLM-based follow-up question generation** This final set of questions aims to increase coverage of video details and generate highly fine-grained questions by leveraging the already collected Q&A pairs for each segment and feed them to an LLM that generates "follow-up" questions that are more detailed and challenging than the initial questions.

```
Follow-up question generation prompt

I have the following information gathered about the video: [list of previous Q&A
samples] Utilizing information and details from all the provided Q&A pairs (make sure
to specialize questions based on the already corrected answers, e.g., using referring
expressions), ask [N] most relevant and interesting, visual questions that we can
ask annotators in order to reveal NEW, rich, additional fine-grained details about
the video that we don't know yet, in particular about the following question types:
'tools/ingredients', 'object counts', 'repetition counts', 'direction of movement',
'hand pose', 'fine-grained object locations', 'spatial relations', 'initial state/end
state', 'action happened before/after', 'clothes wearing', 'body pose', 'main action
in the video', 'temporal extent of action', 'sizes'. The questions should be specific
and have a specific answer. Avoid generic questions that can be very tedious to answer,
e.g., how many objects are there in the scene. Also, do not generate questions that
start with "Is ..." and then list options. Prefer open-ended questions, e.g., starting
with "How". [... More examples & formatting ...]
```

### F.1.3 Automatic Answer Generation

The next step of the data engine aims to produce correct and comprehensive answers to the generated questions. We obtain automatic answers to the generated questions using a version of PLM that has been fine-tuned with extra privileged information of various forms as input. The privileged information includes textual annotations from the metadata available with the candidate training

| Question Type | Sample Questions |
|---|---|
| Action Recognition | What is the process being performed on the sandpaper? 
 What is the action shown? |
| Action Sequence | What does the person do after brewing the tea? 
 What does the person do before marking the vinyl with a pencil? |
| Counting Problems | What is the quantity of universal down cleaner being poured into the task area? 
 How many branches does the person cut in total? 
 How many times does the person spray Greased Lightning onto the ketchup spill? |
| Movement Direction | In what direction is the black welding tool pointing while the person is working on the metal joint? 
 How does the person chop the garlic with the knife? |
| Object Attributes | What is the color of the seatpost shown in the video segment? 
 What is the shape of the tube at the end of the step? 
 What is the size of the knife being used to chop the spring onions? |
| Object Location | Where does the person put the honey bottle away? 
 Where does the person position the clothes before ironing? |
| Object Recognition | What type of roller and paint are being used? 
 What does the person place on top of the smooth half of the egg carton? 
 What was the person initially holding in their left hand? |
| Object State | How would you describe the sink at the beginning of the cleaning process? 
 What is the state of the nematode after mixing it with water and sponge? |
| Other | At what point in the video is the person seen holding the wires? |
| Pose | How are the woman's legs positioned while she is sitting? 
 How bent is the left elbow during the activity? |
| Spatial Relations | How far is the bias tape maker from the right edge of the ironing board? 
 What is the spatial relationship between the bowls and the Brussels sprouts on the kitchen countertop? |
| Speed/Force | How would you describe the consistency of pressure applied during sanding? 
 How fast does the person initially push the stone? |

Table 20: PLM–FGQA question types and sample questions

videos and feature embeddings extracted from off-the-shelf models. Useful textual metadata include the video title, ASR captions or written descriptions, video-level task name (infered by an LLM using the title and captions), and any existing QAs for that video. Off-the-shelf embeddings include frame-level features extracted denseley at 1 fps; we use an open-vocabulary object detection model, OWLv2 [226], for embedding object detection information and CLIP ViT-L14 embeddings [227] for scene classification information. We incorporate the textual annotations directly into language prompts using the following template:

```
Automatic answer generation prompt
A video is showing a task [video level task name], specifically the part where [ASR
caption].  Here is what we already know about the video: [existing question-answer
pairs]. Answer this question in detail: [question]
```

The off-the-shelf embeddings are incorporated into the PLM input via an additional Perceiver-IO[228] tokenizer, which summarizes the embeddings at the segment level.

We fine-tune the answer generator on 1M manually annotated QA pairs. After fine-tuning, we deploy the trained answer generator with privillged information access on the unlabelled questions produced in the previous step, to produce automatic answers.

### F.1.4 Human Annotation

After obtaining segments and generating questions and automatic answers, we employ human annotators to obtain high-quality answers. Our answer annotations include the following:

- **Human-verified answers**: Raters are provided with the model-generated answer and are asked to accept or reject the answer. They can reject questions for being irrelevant or unanswerable, and answers for being factually incorrect or lacking details. Accepted question-answer pairs proceed without changes, while rejected ones are handled differently: question-related rejections (irrelevant or unanswerable) are discarded, whereas answer-related rejections (factually incorrect or lacking details) are marked for correction in the next phase. 17.8% of the total training samples are human-verified automatic answers.

- **Human annotated answers**: Raters answer the questions from scratch by ensuring to cover all the relevant details within the temporal segment. They receive reference information, such as video-level task names and ASR captions, and may use online resources like WikiHow for additional context. Questions that cannot be answered based on the video segment (for example, due to some false premise) are rejected (with an explanation). These manually annotated answers make up 82.2% of the PLM–FGQA training split, and 100% of the evaluation set.

**Quality Control.** Data quality is crucial for model success. We followed several strategies to monitor and enhance annotation quality: *annotation Certification* - we reviewed a small sample of annotations from each rater before they could work in production queues, ensuring that annotators met high-quality standards before advancing to production; *golden Examples* - annotators were provided with high-quality annotation examples, highlighting common error patterns and offering acceptable answers. *targeted and Dual QA* - we conducted daily audits, including vendor auditing and our own sampled quality control. In total, 13% of the training set was audited, and 100% of the samples in PLM–VideoBench underwent quality control.

## F.2 FGQA PLM–VideoBench Construction

| | Train | Test |
|---|---|---|
| **Sources stats** | | |
| Total Videos | 767k | 3.6k |
| Unique Source Videos | 251k | 1.9 |
| Average Duration (sec.) | 9.8 | 12.3 |
| **Annotations stats** | | |
| Number of QA Pairs | 2.4M | 4.2k |
| Number Question Types | 12 | 12 |
| Question Length (avg/max) | 12/114 | 12.3/56 |
| Answer Length (avg/max) | 13.3/911 | 14.1/62 |
| Annotation Type | Human | Human |
| Open-Domain | Yes | Yes |

Table 21: Statistics of the PLM-FGQA training and test data. The test split refers to the FGQA module of PLM–VideoBench.

The FG-QA component of PLM–VideoBench is formed from a held-out portion of PLM–FGQA. We refine this set and transform it into a challenging MCQ-based benchmark by (1) generating MCQs, (2) filtering out samples that can be answered by text-only (blind) LLMs, (3) performing human verification of negatives, and (4) balancing the distribution of question types and domains. The statistics of the dataset are summarized in Table 21. In more detail the steps we followed are:

*MCQ Generation*: To transform QAs into challenging MCQs for evaluation, instead of generating random incorrect answers, we prompt LLMs to produce hard negatives that are semantically close to the correct answer. We use the following prompt which was designed to generate distractors that differ from the correct answer by only a single detail. In effect this enables evaluation to assess fine-grained reasoning about object attributes and tool distinctions.

*Filtering Text-Only Answers*: To ensure that video-based reasoning is required, we test whether a text-only LLM can answer the question correctly without seeing the video. If a question can be answered correctly from text alone, we remove or modify it to emphasize visual and temporal grounding.

*Human Verification of Negatives*: Automatically generated negatives may sometimes be factually true despite being labeled as incorrect. To address this, we perform human verification, where annotators

review distractors to confirm that they are both plausible yet definitively incorrect given the video context.MCQs with ambiguous distractors are removed.

*Balancing Question Types*: Finally, after the above postprocessing and filtering is done, we rebalance the test set, to make sure that the question type and domain distributions are approximately uniform, by undersampling over-represented qyestion types and domains.

**Note on the evaluation metric.** We report the multi-binary accuracy (MBAcc) [39] to evaluate on the FG-QA task. This accuracy is calculated by comparing the correct answer to each distractor individually. Specifically, for each question, we generate a series of binary questions, where the correct answer is compared with one distractor at a time. A prediction is considered correct only if the correct answer is consistently selected across all binary comparisons. We preferred this metric to vanilla MCQ accuracy as it greatly reduces the predictability of automatically-generated MCQs.

```
MCQ generation prompt

Here is a question and answer pair about a video:
Q: [question]
A: [answer]
You need to transform this into a high-quality multiple-choice question. To do this,
first rephrase the given correct answer and then provide n distractor answers. The n
incorrect answers should be reasonable and valid responses to the question, but should
have a different meaning than the correct answer. You generate an incorrect answer
from the correct one by changing a single detail, e.g. an object or verb/action that
is relevant to what's being asked. Make the incorrect answers realistic, plausible
and similar enough to the correct answer so that it is very difficult for someone to
distinguish between them with prior knowledge alone. Finding the correct answer should
also require visual information about the scene. The distractor answers should answer
the question, but should be incorrect but in a non-obvious way. When changing a single
detail to create the distractors, make sure that this detail is the main point of the
question.  For example, if the question is about the color of an object, then the
distractor should change the color of the object and not the kind of object.
Here are some examples of good distractors (desired) and bad distractors (to be
avoided):
Q: What is the person wearing on their hands while applying varnish?
A: The person is wearing white gloves on their hands while applying varnish with a
brush.
Good distractors:
- The person is wearing black gloves on their hands while applying varnish with a brush.
Bad distractors:
- The person is wearing black gloves on their hands while applying paint with a roller.
... More examples & formatting ...
```

# G   PLM–STC Details

We present PLM Spatio-Temporal Captions (PLM–STC), a novel dataset aimed at training and evaluating VLMs for spatial-temporal reasoning. We collected pairs of mask tublets for objects in videos, along with their corresponding detailed temporal descriptions. The annotations are collected on top of the SA-V [57] videos, which are diverse and high-quality. We excluded the test set videos from SA-V, to avoid any data cross contamination. Table 23 provides statistics about the dataset, such as number of total samples, training/val/test splits, object types, and time-segment duration. PLM–STC, is not only novel, but also larger and higher quality compared to existing datasets, see Table 22. In Fig. 5 (right), we show an example of our spatio-temporal captions, describing a little girl (highlighted in **blue**): *(frame 0-81): A little girl moves back as beluga whale approaches her face. (frame 82-85): Out of frame. (frame 86-98): She tries to feed the whale.*

| Dataset | Spatial Type | Year | #Videos | Regions | Temp. Seg. | Captions? |
|---|---|---|---|---|---|---|
| DAVIS16-RVOS [229] | Segmentation | 2018 | 50 | 50 | - | No |
| DAVIS17-RVOS [230] | Segmentation | 2018 | 90 | 205 | - | No |
| YouCook2-BB [47] | BBox | 2018 | 647 | - | 4.3K | No |
| A2D Sentence [231] | Segmentation | 2018 | 3.7K | 4.8K | - | No |
| J-HMDB Sentence [232] | Segmentation | 2018 | 928 | 928 | - | No |
| ActivityNet Entities [233] | BBox | 2019 | 14.3K | 1.5M | 52K | No |
| VidSTG [9] | BBox | 2020 | 6.9K | 44.8K | - | No |
| Refer-Youtube-VOS [234] | Segmentation | 2020 | 3.9K | 7.5K | - | No |
| HC-STVG [235] | BBox | 2021 | 16K | 16K | - | No |
| VLN [56] | Mouse Trace | 2023 | 50K | 43.1K | 43.1K | Yes |
| MeVis [236] | Segmentation | 2023 | 2K | 8.8K | - | No |
| **PLM**-STC | Segmentation | 2025 | 45.7K | 122.3K | 194.2K | Yes |

Table 22: **Spatio-Temporal-Captioning datasets comparison.**

We describe the overall annotation process in Appendix G.1, and how we build the three sub-tasks in Appendix G.2.

## G.1   Annotation Process

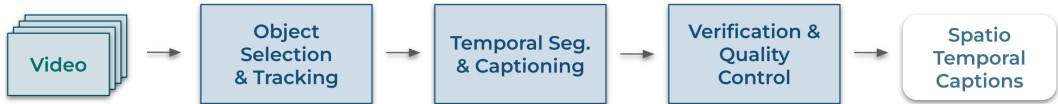

Figure 14: **PLM-STC Annotation pipeline.**

The annotation process is summarized in Figure 14. The annotation process involves three stages: *Object Selection and Tracking*, *Temporal Segmentation and Captioning* and *Verification and Quality Control*.

### G.1.1   Object Selection and Tracking

Annotators select interesting objects with significant motion changes in the video and use SAM 2 [57] to generate initial mask tublets, which they then refine to ensure high-quality spatial-temporal segmentation. We instructed the annotators by defining interesting regions in video footage as those with the presence of significant, dynamic actions performed by subjects, which can be human, animal, or object. These regions involve multiple major actions that evolve over time, rather than static or insignificant actions. We provided annotators with examples of interesting regions, such as one featuring a person making a sandwich, a dog chasing a cat, or a kite getting stuck in a tree. The goal for the annotator is to identify regions with high delta, where the subject performs a sequence of significant activities that change over time, such as a person entering a room, sitting down, and then drinking from a glass. By focusing on these dynamic and evolving actions, annotators can effectively select regions worthy of captioning. Finally, annotators are provided with several examples of good and bad annotations.

### G.1.2 Temporal Segmentation and Captioning

Based on the selected mask tublets, another set of annotators provides time segments for each action and fills in the caption within each time segment. The annotators are instructed to focus on capturing major actions, avoiding minor details or unnecessary movements. When writing captions for each segment, they must ensure clarity in describing the subject's movements and directionality. Additionally, the annotators are advised to avoid making assumptions about the subject's actions or adding details not clearly visible, sticking only to what is directly observable in the frame. As in the previous task, the annotators are provided with several examples of good and bad annotations to guide their work.

### G.1.3 Verification and Quality Control

A final set of annotators manually verifies the tublets and time-segment captions to ensure accuracy and consistency. For mask refinement, we re-run the same pipeline as §G.1.1, while not letting the annotators choose the interesting object, but only refine the quality of the mask. For captioning refinement, the annotators are tasked with three objectives: 1) Redundancy: eliminate any repeating or redundant information to ensure the caption is concise; 2) *Accuracy*: verify that every word in the caption accurately describes a fact present in the video, correcting or removing any incorrect information; and 3) *Actions*: add missing major action information to the caption while preserving existing atomic actions, ensuring the caption effectively conveys the key events in the video.

### G.2 PLM–STC Benchmark

|  | All | Train | Val | Test |
|---|---|---|---|---|
| **Dataset stats** | | | | |
| Number of Videos | 45.2K | 42.0K | 804 | 2.3K |
| Spatio Temporal Caption | 127.8K | - | - | - |
| Temporal Caption | 198.7K | - | - | - |
| **Tube's categories** | | | | |
| Person | 104.5K | 99.6K | 861 | 2.4K |
| Animal | 16.8K | 13.2K | 550 | 1.7K |
| Object/things | 6.4K | 4.4K | 436 | 1.2K |
| **Temporal captions per Tube** | | | | |
| 1 caption per tube | 78.9K | 73.9K | 842 | 2.4K |
| 2 caption per tube | 30.9K | 27.8K | 566 | 1.7K |
| 3 or more Caption per tube | 16.38K | 14.15K | 421 | 1.2K |
| **Tasks stats** | | | | |
| Region Detailed Captioning (RDCap) | 122.3K | 117.2K | 2.5K | 2.6K |
| Region Captioning (RCap) | 194.2K | 179.5K | 4.6K | 10.1K |
| Region Temporal Localization (RTLoc) | 192.0K | 179.5K | 4.6K | 7.9K |

Table 23: PLM–STC dataset statistics. Note the for RTLoc, we filter the test set to include only the captions that are unambiguously localized, *i.e.*, they map to a single time window in the video. As a result, the test set size is reduced to 7,910 instances compared to RCap.

We utilize the collected data to train and evaluate the PLM on three challenging tasks that are essential for video perception. Firstly, we created a balanced validation and test split by the combination of tube categories and number of caption per tube while making sure no video overlaps with the training set. This is done to make sure we evaluate all the categories presents in the dataset equally. Then, we process the data for each task:

**Dense Video Region Captioning (RDCap).** This comprehensive task combines both "what" and "when" aspects. The model takes the video and the tubelets as input and outputs the full time-segment captions. We also assign an *out of frame* caption to temporal segments for which the subject does not appear in the video to ensure dense temporal coverage of events across the video duration.

**Video Region Captioning (RCap).** This task involves describing "what" activities are performed within a specific time frame by the objects in the tubelets. The model receives the video, the tubelets, and the temporal region as input and outputs the corresponding captions. We filter out events that refer to the subject when it is out-of-frame to avoid evaluating trivial captions.

**Region Temporal Localization (RTLoc).** This task requires the model to localize "when" specific events occur in relation to a given tubelet. The input includes the video, the tubelet, and the caption, while the output is the start and end frames indicating when the captioned event occurs. Like RCap, we filter out out-of-frame events, as well as ambiguous events that may be be localized to multiple time segments. For example, if the subject opens the door twice, the event text are guaranteed to be unique (*e.g.*, referring to the first and second time they opened the door) or dropped entirely if ambiguous (*e.g.*, if the text only mentions the action).

These tasks are designed to both improve and evaluate the model's capabilities, with the same input-output format applied during both training and evaluation. Figure 6 illustrate an examples of the task, including the prompt used to train and evaluate the PLM.

# H  Smart Glasses Data

## H.1  Data collection and annotation

We collected the source videos for PLM-SGQA using commercial smart glasses, which enable participants to capture egocentric videos in a hands-free manner. Participants are presented with 14 categories of popular scenarios, such as shopping, cooking, and walking in a neighborhood, and are instructed to ask questions about their surroundings as if interacting with a multi-modal assistant that shares their visual perspective. Specifically, participants are asked to ask questions spontaneously, without delay, about the things they see and experience, and to focus on visual queries rather than dynamic information that may change regularly. After recording the videos, participants annotate the segments by marking the start and end points of the video relevant to each question, as well as providing the ground-truth answer.

## H.2  SGQA Benchmark

To create the SGQA component of PLM–VideoBench we first filtered the Q&As using an LLM to obtain a shortlist of questions that focus on human activity and also are perception-based rather than based on general knowledge. This means that SGQA focus on questions that require good visual understanding of the scene to be accurately answered. This process yields an evaluation set consisting of 655 Q&As. For the resulting Q&As, we then trimmed the original videos to obtain clips within the temporal boundary that the human wearer/annotator specified. As the annotated segments end at the point where the smart-glass wearer asks the question, it is important for all evaluations to specify that the question refers to the end of the video clip – *e.g.* see the prompt we used for PLM and baselines evaluation in 11. We summarize the statistics of the SGQA test set in Figures 15 and 16.

| Sources stats | |
|---|---|
| Total Videos | 663 |
| Average Duration (sec.) | 29.4 |
| **Annotations stats** | |
| Number of QA Pairs | 665 |
| Number Domains | 14 |
| Question Length (avg/max) | 9.0 / 52 |
| Answer Length (avg/max) | 21.6 / 40 |
| Annotation Type | Human |
| Open-Domain | Yes |

Figure 15: Statistics of the PLM-SGQA test data.

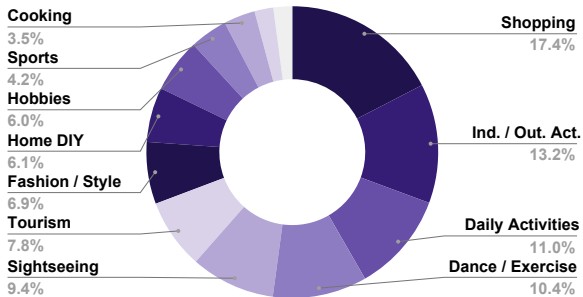

Figure 16: Domain distribution of video-clips in PLM-SGQA.

# I Synthetic Data Engine

Our data engine targets *base* capabilities of VLMs: image captioning, visual question answering, OCR, chart/diagram understanding, and video understanding. We developed different pipelines for images and videos, and includes different levels of metadata to generate captions and QAs.

**Image Captions**: We caption high-quality images using Llama 3.1V 90B. An example is shown in Figure 17. We use this pipeline to caption SA1B [14], Object365 [135], and OpenImages [136].

**OCR QAs**: We leverage pre-extracted OCR and use it as input for a LLM (*i.e.*, Llama 3.3 70B) to generate a set of five question-answer pairs. An example is shown in Figure 18. We use this pipeline to generate QAs for PDFAcc [132], and UCSF [133].

**Image Captioning plus QAs**: In cases for which OCR does not provide enough information to create questions (*e.g.*, scientific figures), we futher caption the image using Llama 3.1V 90B. Then we pass the caption with auxiliary metadata (*e.g.*, OCR) to a LLM (*i.e.*, Llama 3.3 70B) to generate question-answers pairs. An example is shown in Figure 19). We use this pipeline to generate captions and QAs for ArxivQA [104], DocVQA [26], InfoVQA [27] and Ai2d [67].

**Video Captioning plus QAs**: An image captioner is run on key-frames of the video, as well as a video captioner on the overall video at 1 fps. The result captions are passed to a LLM (*i.e.*, Llama 3.3 70B, or Llama 3 405B) with additional metadata (*e.g.*, video title etc.), so to generate a detailed caption and a multiple-choices question answers pair. An example is shown in Figure 20). We use this pipeline to generate captions and QAs for YT-1B [237], Ego4d [43] [4], DiDeMo [140], Charades [139], and Kinetics710 [54] [5].

---

### Detailed Caption

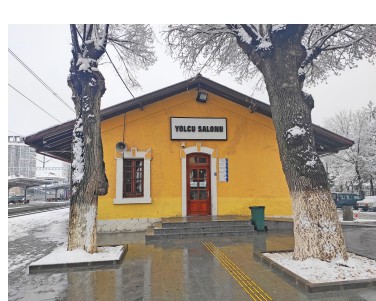

The image depicts a yellow building with a red door and a sign that reads "YOLCU SALONU" in black letters. The building has a gable roof and a small window on the left side. There are two trees in front of the building, one on each side of the entrance. The ground is wet, and there is snow on the trees and the ground. In the background, there is a train station with a train on the tracks. The sky is gray and overcast. The image appears to be a photograph taken on a cold winter day. The presence of snow and the gray sky suggest that it is a winter scene. The building and the train station in the background suggest that it is a public place, possibly a train station or a bus stop. The sign on the building reads "YOLCU SALONU," which is Turkish for "Passenger Lounge." This suggests that the building is a waiting area for passengers. Overall, the image depicts a quiet and peaceful winter scene, with a focus on the building and the surrounding environment.

---

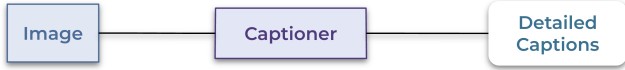

Figure 17: Detailed caption example, and the corresponding pipeline. The captioner (*i.e.*, Llama 3V 90B) is prompted to generate the caption for the provided image.

---

[4]For this dataset we used Llama3 405B, rather than Llama 3.3 70B

[5]DiDeMo, Charades, Kinetics710 used a simpler pipeline where only frame captions were used, and a smaller scale LLM (*i.e.*, Llama 3.1 8B)

## OCR

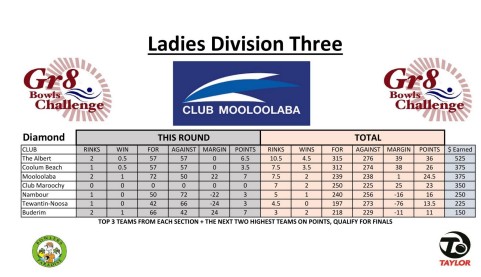

Ladies Division Three THIS ROUND TOTAL Diamond CLUB RINKS WIN FOR AGAINST MARGIN POINTS RINKS WINS FOR AGAINST MARGIN POINTS Earned The Albert 2 0.5 57 57 0 6.5 10.5 4.5 315 276 39 36 525 Coolum Beach 1 0.5 57 57 0 3.5 7.5 3.5 312 274 38 26 375 Mooloolaba 2 1 72 50 22 7 7.5 2 239 238 1 24.5 375 Club Maroochy 0 0 0 0 0 0 7 2 250 225 25 23 350 Nambour 1 0 50 72 -22 3 5 1 240 256 -16 16 250 Tewantin-Noosa 1 0 42 66 -24 3 4.5 0 197 273 -76 13.5 225 Buderim 2 1 66 42 24 7 3 2 218 229 -11 11 150 TOP 3 TEAMS FROM EACH SECTION + THE NEXT TWO HIGHEST TEAMS ON POINTS, QUALIFY FOR FINALS

## Question Answers

Q: How many teams qualify for the finals?
A: The top 3 teams from each section, plus the next two highest teams on points, qualify for the finals, making a total of 5 teams.
Q: How many points has the Buderim team earned?
A: The Buderim team has earned 150 points.
Q: Which team is currently in the lead?
A: The Albert team is currently in the lead with 36 points and a margin of 39.
Q: What is the current margin of the Mooloolaba team?
A: The Mooloolaba team has a current margin of 1.
Q: What is the overall summary of the document?
A: The document appears to be a standings table for the Ladies Division Three of a bowling or lawn bowls competition, showing the current standings of various teams, including their win-loss records, points, and margins.

## Pipeline

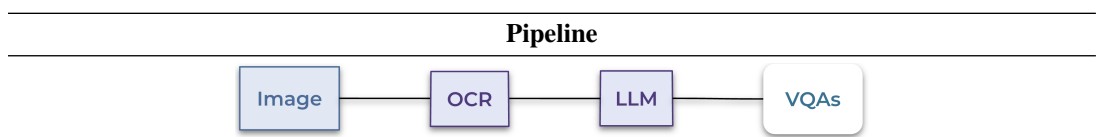

Figure 18: Visual Question Answering pairs and the corresponding pipeline. The OCR text is extracted from the image, and passed to the LLM (*i.e.*, Llama 3.3 70B) to generate QA pairs.

## OCR

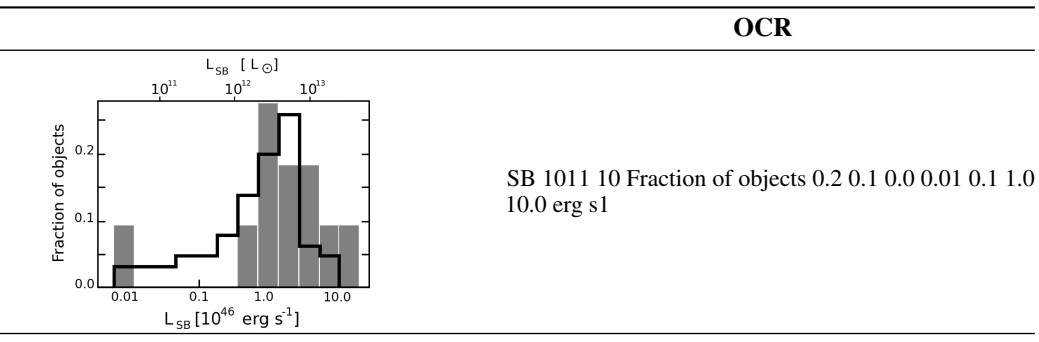

SB 1011 10 Fraction of objects 0.2 0.1 0.0 0.01 0.1 1.0 10.0 erg s1

## Detailed Caption

The image depicts a histogram of the distribution of objects, with the x-axis labeled "$L_{SB}[10^{46}ergs^{-1}]$" and the y-axis labeled "Fraction of objects." The x-axis ranges from 0.01 to 10.0, while the y-axis ranges from 0.0 to 0.2. The histogram is divided into bins of varying widths, with the first bin spanning from 0.01 to 0.1, the second bin spanning from 0.1 to 1.0, and so on. Each bin contains a bar representing the fraction of objects within that range. The bars are shaded in gray, with some bins having multiple bars. A key feature of the histogram is the presence of a peak in the middle bin, which corresponds to an $L_{SB}$ value of around 1.0. This suggests that the majority of objects have an $L_{SB}$ value close to this value. The histogram also shows a tail extending towards higher $L_{SB}$ values, indicating that some objects have significantly higher $L_{SB}$ values than the majority. Overall, the histogram provides a visual representation of the distribution of $L_{SB}$ values among the objects being studied. It allows for easy identification of patterns and trends in the data, such as the peak in the middle bin and the tail towards higher $L_{SB}$ values.

## Multi-Choice Question Answer (MCQA)

What is the approximate $L_{SB}$ value at which the majority of objects have their peak?
Options:
(A) 0.1
(B) 1.0
(C) 5.0
(D) 10.0
Answer: (B) 1.0.

## Pipeline

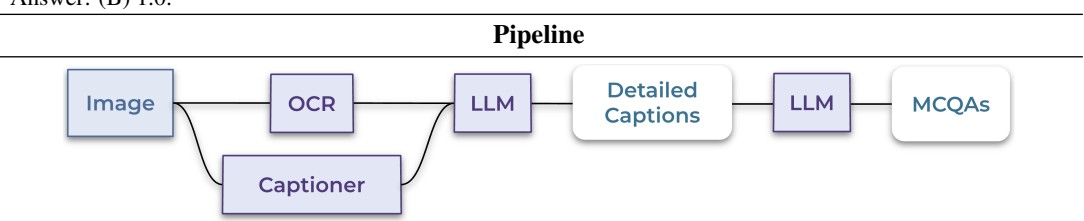

Figure 19: Detailed Captions and Multi-Choice Question Answers (MCQAs) and the corresponding pipeline. The OCR text is extracted from the image, and the caption is generated by the captioner (*i.e.*, Llama 3V 90B), which are all passed to the LLM (*i.e.*, Llama 3.3 70B) to generate MCQAs.

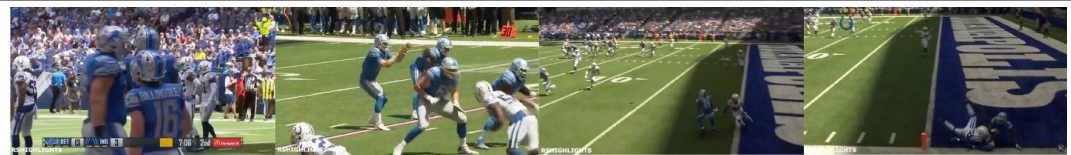

**Metadata**

**Title**: Lions VS Colts Highlights 2017 Preseason Game
**Description**: Comment suggestions for future videos and Enjoy!

**Frame Caption**

Frame 435: The image shows a man with dreadlocks standing in front of a crowd of people in a stadium. He is wearing a white t-shirt and is surrounded by a group of people standing on the ground. On the left side of the image, there is a table fan, bottles, and other objects placed on a table. In the background, there are people sitting on chairs, stairs, railings, boards with text, lights, and the sky. The text on the boards reads "Indianapolis Colts vs San Francisco 49ers

Frame 585: The image shows a football game being played on a TV screen, with a group of people on the ground and a few people standing in the background. At the bottom of the image, there is text and numbers indicating that the game is between the Indianapolis Colts and the Detroit Lions.

Frame 765: The image shows a group of people playing a game of football on a green field, with white lines marking the boundaries of the field. At the bottom of the image, there is text and numbers indicating that the game is between the Indianapolis Colts and the Detroit Lions.

Frame 945: The image shows a football game being played on a TV screen, with people wearing helmets and playing on the ground. At the bottom of the image, there is text and numbers indicating that the game is between the Detroit Lions and the Indianapolis Colts.

**Video Caption**

Football players wearing helmets, Detroit Lions vs Indianapolis Colts, player running with ball, falls down, touchdown scored.

**Detailed Caption**

A football player is running with the ball and then falls down, the game is between the Detroit Lions and the Indianapolis Colts, with players wearing blue and white uniforms and helmets, and people sitting in the stadium, a watermark on the image shows the teams playing, one player is laying on the ground next to another player wearing a white and blue jersey and a white helmet, a horseshoe-shaped object is on the grass, text on the image indicates a touchdown, players are wearing helmets and playing on the ground, with watermarks in the bottom left corner of the images, the game involves the Detroit Lions and the Indianapolis Colts, and also shows the Colts playing against the New England Patriots and the Lions playing against the Dallas Cowboys.

**Multi-Choice Question Answer (MCQA)**

What is the outcome of the football player's action?
Options:
(A) the player runs out of bounds
(B) the player scores a touchdown
(C) the player drops the ball
(D) the player falls down
(E) the player gets injured
Answer: (B) the player scores a touchdown.

**Pipeline**

Figure 20: Detailed Captions and Multi-Choice Question Answers (MCQAs) and the corresponding pipeline. An LLM (*i.e.*, Llama 3.3 70B) with a video caption, frame captions at 1 fps, and additional metadata (*e.g.*, video title etc.), generates a detailed caption and multiple-choices question answers pair.

# J Qualitative Results

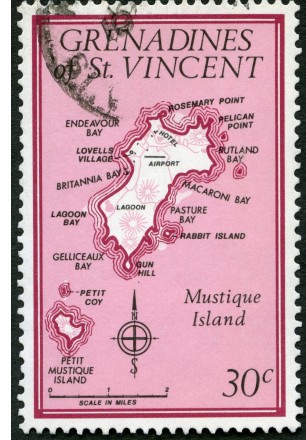

**Question:** Provide a detailed description of the image.
**Answer:** The image is a pink postage stamp featuring a map of the Grenadines of St. Vincent, with a red border and black text. The map shows various locations on the island, including hotels, bays, and other points of interest.
\*Key Features:\*

\* \*Map:\* A pink map of the Grenadines of St. Vincent, with a red border and black text.
\* \*Locations:\* Various locations on the island are marked, including hotels, bays, and other points of interest.
\* \*Scale:\* A scale in miles is provided at the bottom of the map.
\* \*Denomination:\* The denomination "30c" is printed in the bottom-right corner of the stamp.
\* \*Border:\* The stamp has a red border with a serrated edge.

\*Overall Impression:\*

The image appears to be a vintage postage stamp from the Grenadines of St. Vincent, featuring a map of the island and various locations of interest. The stamp has a classic design and is likely from the mid-20th century.

Figure 21: PLM-8B OCR examples.

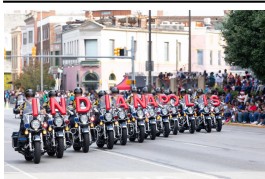

**Question:** What does the word say together from the bikers?
**Answer:** The word says Indianapolis.

Figure 22: PLM-8B OCR in the wild examples.

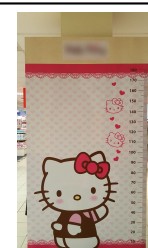

**Question:** How tall is the kitty character in the image?
**Answer:** The kitty character is approximately 100 cm tall.

Figure 23: PLM-8B OCR in the wild examples.

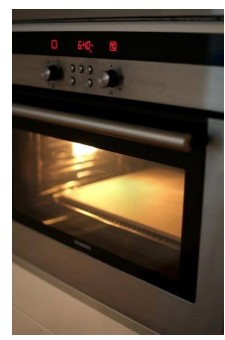

**Question:** What is inside the oven?
**Answer:** There is a baking sheet inside the oven.
**Question:** What temperature is the oven set to?
**Answer:** The oven is set to 200 degrees.
**Question:** What unit is the oven's temperature set to?
**Answer:** The oven's temperature is set to Celsius.

Figure 24: PLM-8B Natural image QA

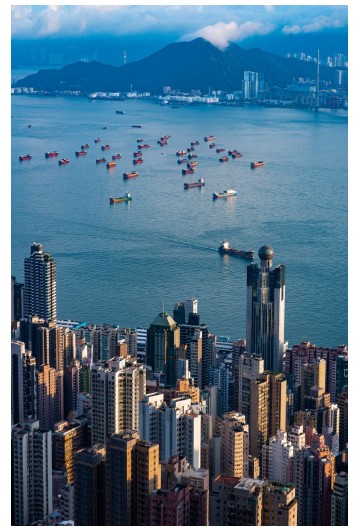

**Question:** Describe the image in details
**Answer:** A city skyline with a large body of water in the foreground. The city skyline is made up of tall buildings, some of which are skyscrapers. The buildings are densely packed together and vary in height and color. In the foreground, there is a large body of water that appears to be a harbor or bay. There are many boats and ships in the water, some of which are docked at the shore while others are sailing or anchored in the middle of the water.

Figure 25: PLM-8B Captioning.

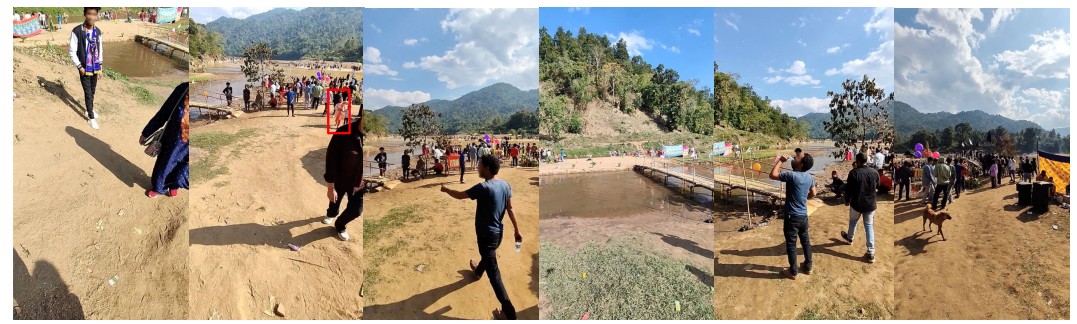

**RDCap**

**Question:** Describe the subject's actions within the red rectangles, providing start and end frames ids and a brief description for each action in a dense caption format. For each item use the format [start, end]: [description] separated by a newline, where start and end are frame numbers between 0 and 31 in this 32 frame video.
**Answer:** Frame [0, 6]: Out of frame
Frame [6, 15]: A woman is walking towards the camera with a child. She is holding the child's hand with her right hand and carrying a bag in her left hand
Frame [16, 31]: Out of frame.

**RCap**

**Question:** What is happening to the subject in the region highlighted by the red rectangle during frames [6, 15] in this 32 frame video?
**Answer:** a woman is walking on a muddy road with a child. she is holding the child's hand with her right hand and adjusting her saree with her left hand while walking

**RTLoc**

**Question:** Please provide the start and end frame of when 'an individual standing near a water body is drinking a from a water from a bottle' happens corresponding to the region marked by the blue rectangle. Use the format [start, end] where start and end are frame numbers between 0 and 31 in this 32 frame video.
**Answer:** [23, 26]

**Captioning**

**Question:** Please describe the video in detail, focusing on actions.
**Answer:** This video features a group of people walking around a muddy area with a small bridge in the background. The video is taken outdoors and the weather is sunny. The video starts with a man standing on the left side of the frame, facing the camera. He is wearing a white shirt, a black vest, black trousers and white shoes. He has a scarf around his neck. The camera pans to the right and shows a group of people walking around. The video is taken from a low angle and the camera is handheld.

The video ends with a man walking towards the left side of the frame. He is wearing a blue shirt, black trousers and brown shoes. He is holding a bottle in his right hand and is drinking from it.

**Fine-grained QA**

**Question:** Please explain what the man in blue t-shirt is pointing at in the video.
**Answer:** The man in blue t-shirt is pointing at the bridge.

Figure 26: PLM-8B Video Region Captioning, Grounding and Fine-grained QA.

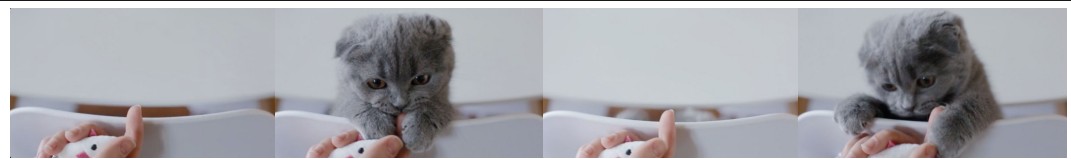

**Question:** Please give me a details of the video focusing on actions and interactions.
**Answer:** This video features a close up shot of a hand holding a plush toy mouse with a grey kitten standing on a white chair, reaching out and touching the toy with its paws. In the beginning of the video, the kitten is seen standing on the chair and reaching out with both paws to touch the toy mouse. The kitten then withdraws its paws and sits back on the chair. The hand holding the toy mouse is seen moving the toy closer to the kitten and then away from the kitten. The kitten then reaches out again and touches the toy with its paws. The hand then moves the toy away from the kitten again and the kitten looks to the left of the frame.

This video is set against a white background.

Figure 27: PLM-8B Video Captioning

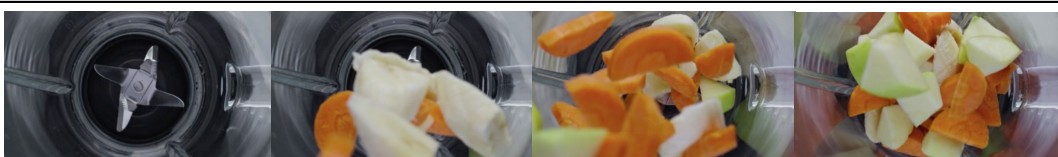

**Question:** What are the ingredients being thrown into the mixer?
**Answer:** Carrots, apples, and bananas.

Figure 28: PLM-8B Video QA

# K   Limitations and Future Work

Our PLM models achieve strong performance against open-data baselines and proprietary models alike, however there is still room for improvement in both modeling and data. On the model front, we do not experiment extensively with long video modeling components (*e.g.*, token compression, dynamic temporal resolution). As a result, our performance on long video benchmarks [41, 123, 125] is less competitive (see Table E). PLM is compatible with such newer advancements and can be incorporated in future work.

Additionally, our results are sensitive to the characteristics of the base LLM. We see especially low performance of PLM on benchmarks such as MMMU [68], MME [95] and Video-MME [20] (see Tables 3 and 4), where the strongest baselines often rely on LLMs that are more verbose, but also have a likely much larger language component (see the gap to proprietary models on some benchmarks). We also note that our model performs relatively poorly on our SGQA task (Table 5), targeting a mix of perception and knowledge based questions to smart glasses. Strong chatbot-focused systems like GPT-4o excel at tasks that go beyond core perception.

On the data front, our mix focuses squarely on visual perception — it does not include for example, multi-step reasoning, robotics or world-knowledge data. Despite these limitations, PLM contributes new capabilities and strong benchmark results, and set a new standard for fully reproducible VLMs.

# L   Broader Impact

Our work aims to advance open and reproducible research in vision-language modeling by releasing models, data, and benchmarks that support open research. By not having any distillation from proprietary models, we hope to improve reproducible and transparent training and evaluation of VLM research. However, like all MLLMs, our Perception Language Model (PLM) may have some risks. Even by carefully selecting datasets and apply several mitigation (CSAM, NSFW, etc.), the model may still contain hidden biases or generate inappropriate or harmful content. We took steps to reduce these risks by teaching the model to refuse answering questions related to bias, harassment, or adult content. We also remove all samples containing any mention of human faces from all the datasets.

We also annotate and release a large-scale dataset for fine-grained video question answering and spatio-temporal grounding. This release has the potential to significantly advance research in image and video understanding. Making the dataset openly available allows others to reproduce our work and invites broader community involvement. This transparency supports safer and more accountable progress, helping researchers better understand and address potential biases or limitations.

We believe that by openly sharing our models and data, while actively addressing ethical concerns, our work can contribute positively to vision-language research.

