# OpenReview forum: "PerceptionLM: Open-Access Data and Models for Detailed Visual Understanding"
_NeurIPS.cc/2025/Conference — NeurIPS 2025 spotlight_

### Official Review · Reviewer_e1P7 · 2025-07-01

**Clarity:** 3
**Significance:** 3
**Originality:** 3
**Rating:** 5
**Confidence:** 4

**Summary:**

This paper addresses the issue of non-transparency in proprietary vision-language models such as GPT-4o, which makes it difficult for the academic community to accurately assess the true capabilities of current models. To overcome this limitation, the authors propose a data engine designed to generate synthetic multimodal data in a scalable yet transparent manner. Using the generated data, they conduct a series of scaling experiments and observe that while synthetic data is effective for general video understanding, it falls short on more challenging video comprehension tasks. To mitigate this, the paper introduces a high-quality video dataset annotated by humans, along with PLM-VideoBench, a benchmark derived from the human-annotated dataset. With a carefully designed training pipeline and data engine, the resulting model family - PLMs - achieves strong performance across a wide range of image and video benchmarks.

**Questions:**

1. Will the data engine be open-sourced or not?
2. Can the limitation that synthetic data does not significantly improve performance on hard video QA tasks be mitigated by using a stronger VLM or LLM?

**Ethical Concerns:**

["NO or VERY MINOR ethics concerns only"]

**Final Justification:**

The rebuttal resolves most of my concerns; accordingly, I maintain my original rating and recommend acceptance.

**Limitations:**

yes

**Paper Formatting Concerns:**

- In Table 3, some best results are not properly highlighted. Eg, Qwen2VL-2B’s performance on VSR benchmark.

**Quality:**

3

**Strengths And Weaknesses:**

### Strengths

1. This paper is well-motivated. Currently, the widespread use of proprietary models as data annotators makes it difficult to study the influence of other contributing factors. A fully transparent and controllable synthetic data engine can significantly benefit research in this area.
2. The dataset and benchmark contributions are both substantial and solid. The inclusion of both synthetic data and human-annotated data provides valuable resources for advancing VLM development.
3. The paper conducts extensive experiments on the scaling behavior of synthetic data and finds that while synthetic data benefits general video understanding, it is less effective for more challenging video QA tasks.

### Weakness

1. Although the paper emphasizes transparency, it adopts LLaMA3 as an open-source alternative to closed-source models. However, LLaMA3’s training process lacks full transparency, and it may have been trained on content generated by proprietary models. This could implicitly influence the quality of the synthetic data and affect PLM performance.
2. Regarding model performance on image benchmarks, PLMs exhibit strong OCR capabilities but perform poorly on knowledge-intensive tasks such as MMMU. This raises the question of whether such disparities stem from biases in the data engine.
3. A similar pattern is observed in video benchmarks: PLMs perform well on PerceptionTest but fall significantly behind on VideoMME. This inconsistency warrants further investigation.
4. Since PLM-VideoBench is an in-distribution byproduct of the high-quality human-annotated dataset, the model’s performance advantage on this benchmark over other models is less compelling.

---

> ### Author Rebuttal · Authors · 2025-07-30
>
> We thank the reviewer for recognizing the key contributions of our work. We appreciate the acknowledgment of the motivation behind addressing the non-transparency of proprietary vision-language models such as GPT-4o. We also appreciate the recognition of our transparent and controllable synthetic data engine, the release of both synthetic and human-annotated datasets, and the scaling analysis that highlights the limitations of synthetic data on challenging video reasoning tasks.
>
> ---
>
> > 1. Although the paper emphasizes transparency, it adopts LLaMA3 as an open-source alternative to closed-source models. However, LLaMA3’s training process lacks full transparency, and it may have been trained on content generated by proprietary models. This could implicitly influence the quality of the synthetic data and affect PLM performance.
>
> We thank the reviewer for the observation. In our work, we differentiate between closed-weight proprietary models (e.g., GPT-4, Gemini) and open-weight models such as LLaMA3. While LLaMA3’s full training corpus is not publicly available, its model weights, training setup, and outputs are accessible and reproducible, unlike closed models whose generations cannot be independently verified. We select LLaMA3 specifically to ensure transparency and reproducibility in our synthetic data engine.
>
> We understand the reviewer’s concern that the full data source of LLaMA training is not disclosed. We have contacted the authors of LLaMA3 and confirmed that no artifacts of proprietary models (e.g., contents generated by closed models) was used in training LLaMA3.
>
> ---
>
> > 2. Regarding model performance on image benchmarks, PLMs exhibit strong OCR capabilities but perform poorly on knowledge-intensive tasks such as MMMU. This raises the question of whether such disparities stem from biases in the data engine.
>
> > 3. A similar pattern is observed in video benchmarks: PLMs perform well on PerceptionTest but fall significantly behind on VideoMME. This inconsistency warrants further investigation.
>
> We thank the reviewer for the observation. We hypothesize that some benchmarks such as MMMU and VideoMME rely heavily on the capabilities of the underlying language model. To test this, we conducted a separate ablation experiment comparing LLaMA3 and other popular LLM.
>
> Specifically, we extract the vision encoder from PLM after Stage 2, and perform a short warm-up (Stage 1) followed by a shorter version of SFT (Stage 3) using newly initialized LLM, LLaMA3.1-Instruct 8B and Qwen2.5-Instruct 7B models. Beside the LLMs, everything else is controlled (training data, # iterations, random seed, etc.)
>
> From the result, we observe that results are consistently better with Qwen2.5: VideoMME improves from 48.0 to 54.1, and MMMU from 39.9 to 48.1. These results suggest that the disparities on knowledge-intensive tasks are more likely due to differences in LLM capabilities rather than biases in the data engine. We have also noted this in the limitations section (Appendix L, Line 1619) of the paper.
>
> ---
>
> > 4. Since PLM-VideoBench is an in-distribution byproduct of the high-quality human-annotated dataset, the model’s performance advantage on this benchmark over other models is less compelling.
>
> We thank the reviewer for the insightful comment on the comparison of PLM with previous methods where PLM included in-distribution data, and the other methods did not. To mitigate this, we conduct an experiment by training PLM after removing human annotated in-distributed data. The table below shows the results. We notice that PLM still performs better than Qwen2.5-VL and InternVL2.5 models, indicating that PLM performs better in a fair setting where no in-domain data is included in the training.
>
> | Model              | FGQA | SGQA | RDCap | RCap | RTL  |
> |--------------------|------|------|-------|------|------|
> | Qwen2.5 VL 3B      | 43.7 | 45.1 | 0.3   | 17.2 | 13.9 |
> | InternVL2.5 4B     | 50.0 | **49.2** | 4.9   | 25.9 | 15.4 |
> | **PLM 3B no hum.** | **54.0** | 41.5 | **11.1**  | **27.6** | **25.0** |
> | PLM 3B         | 67.1 | 38.8 | 53.1  | 45.0 | 58.2 |
>
> ---
>
> > 1. Will the data engine be open-sourced or not?
>
> Yes, the data engine will be open-sourced along with the datasets, models, and training recipes to support transparency, reproducibility, and further research in the community.
>
> ---
>
> > 2. Can the limitation that synthetic data does not significantly improve performance on hard video QA tasks be mitigated by using a stronger VLM or LLM?
>
> We thank the reviewer for the insightful question. During the development of PLM, LLaMA3 and LLaMA3V were the best language and vision-language models that were not trained with any content generated by proprietary models. Molmo used Claude for re-captioning, while QwenVLs and InternVLs used GPT-generated data. During development, we tried using LLaMA3.1 405B for our data engine, but the results were similar compared to LLaMA3.3 70B, despite a much higher inference cost. Our data engine was designed after multiple iterations and optimized for both scalability and accuracy.
>
> Further, regarding the parameter count in PLM, in Figure 3, we show results across three model scales, 1B, 3B, and 8B, and observe the same trend: scaling synthetic data improves performance on general video understanding tasks, but shows limited gains on Hard Video QA tasks. While larger models (e.g., 8B) generally performs better than smaller ones (e.g., 3B), increasing the amount of synthetic data for each does not significantly improve performance on Hard QA (i.e. Cg-Bench, Tomato, EventHallusion, MotionBench, TemporalBench, LVBench, and Charades-STA). These tasks require fine-grained visual understanding over time, and we believe that further increasing the LLM size (e.g., to 70B or even 405B) would still exhibit similar limitations, as the bottleneck lies in visual reasoning rather than language modeling capacity.
>
> However, as shown in Figure 5, adding PLM human-annotated data, which provides rich video-level fine-grained QAs and spatio-temporal grounded captions, leads to significant improvements on Hard QA. This supports our view that stronger visual supervision, not larger language models, is key to addressing the limitations in these benchmarks.

---

> > ### Comment · Reviewer_e1P7 · 2025-08-03
> > **Response to Authors' Rebuttal**
> >
> > Thank you for your detailed response, most of my concerns have been addressed. Regarding Weakness 1 in my review, I would first like to express my appreciation for your efforts in verifying LLaMA3’s training recipe. However, I would still like to raise a point: given the widespread use of ChatGPT-like language models across the internet, it is increasingly difficult to guarantee that all web-crawled data are entirely free from LLM-generated content. Therefore, I encourage the authors to briefly and appropriately discuss the potential presence of AI-generated content in the base model or training data, and to raise awareness of this issue in the paper.
> > Thank you again for your hard work.

---

> > > ### Author Response · Authors · 2025-08-04
> > >
> > > We are glad that most of the reviewer's concerns have been addressed. Regarding the remaining concern (Weakness 1), we appreciate the reviewer's suggestion and agree that this issue deserves explicit mention in the paper to raise awareness. We will add a discussion on the potential risks of AI-generated content present in large data corpora.

---

### Official Review · Reviewer_s99d · 2025-07-01

**Clarity:** 3
**Significance:** 3
**Originality:** 4
**Rating:** 6
**Confidence:** 3

**Summary:**

The main contributions of this work are embodied in its transparent multimodal data construction pipeline, which incorporates both synthetic data and manually annotated data. By exploring scaling laws with synthetic data, this study reveals that even extensive synthetic datasets struggle to improve Vision-Language Models' performance on challenging tasks and novel scenarios, thereby underscoring the critical importance of high-quality human-annotated data.

The contributions of this work include:
1. Transparent synthetic data pipelines
2. Open-access datasets and models
3. Enhanced reproducibility supported by comprehensive experimental analysis and systematic interpretation.

**Questions:**

1. Does the paper provide a clear explanation regarding "the significance of each training stage"?
2. Will other synthetic data (66.1M) besides the high-quality human-annotated video data also be open-sourced?

**Ethical Concerns:**

["NO or VERY MINOR ethics concerns only"]

**Final Justification:**

All concerns have been addressed, and I am willing to raise my score.

**Limitations:**

yes

**Paper Formatting Concerns:**

No formatting concerns.

**Quality:**

3

**Strengths And Weaknesses:**

Strengths:
1. The transparent synthetic data construction pipeline, open-access high-quality human-annotated data, and transparent training strategies are highly valuable for the VLM research community.
2. PLM–VideoBench addresses a critical gap in the research community by providing a benchmark for evaluating VLMs' holistic video understanding capabilities.
3. Extensive experiments demonstrate that these transparent pipelines and datasets enable reproducible performance comparable to both proprietary models and open-weight models.
4. The response regarding "data gaps" is sufficiently addressed.

Weaknesses:

My only minor concern pertains to the discussion of "the significance of each training stage" mentioned in line 30: While I can find some relevant explanations in lines 73-92 and Appendix A.1, these sections primarily focus on implementation details and training settings. Therefore, it remains unclear to me what "the significance of each training stage" specifically refers to, and the authors' systematic exposition of this question appears insufficient.

---

> ### Author Rebuttal · Authors · 2025-07-31
>
> We thank the reviewer for acknowledging the main contributions of our work, including the transparent synthetic data pipeline, open-access human-annotated datasets, and the comprehensive training and evaluation framework. We appreciate the recognition of PLM-VideoBench as a valuable benchmark for holistic video understanding, as well as the usefulness of our systematic analysis in identifying and addressing data gaps in existing VLM training datasets.
>
> ---
>
> > 1. My only minor concern pertains to the discussion of "the significance of each training stage" mentioned in line 30: While I can find some relevant explanations in lines 73-92 and Appendix A.1, these sections primarily focus on implementation details and training settings. Therefore, it remains unclear to me what "the significance of each training stage" specifically refers to, and the authors' systematic exposition of this question appears insufficient.
>
> > 1. Does the paper provide a clear explanation regarding "the significance of each training stage"?
>
> We thank the reviewer for pointing out the need for a clearer explanation of the significance of each training stage. Table 6 provides a controlled comparison in its first two rows that shows the impact of Stage 2. In the first row, only stage 1 + stage 3 is performed, serving as a baseline. In the second row, synthetic data (PLM-Synth ✓) is included, corresponding to Stage 2 of our training pipeline. Importantly, PLM-STC and PLM-FGQA are not included in either of these two configurations. This setup shows that introducing Stage 2 leads to consistent improvements across nearly all metrics.
>
>  Stage 2 | Total Avg | PLM-FGQA (MBAcc) | PLM-SGQA (acc) | PLM-ST (3-metric avg.) | Fine-Grained QA | Video Cap (Dream 1K) | Video QA | Video Hallu | Spatio & Temp | Image OCR | Image Cap | Image Rec |
> | --------- | --------- | ---------------- | --------------- | ---------------------- | --------------- | -------------------- | -------- | ----------- | ------------- | --------- | --------- | --------- |
> | ✗         | 48.5      | 39.7             | 34.4            | 6.6                    | 42.2            | 24.0                 | 67.5     | 64.9        | 50.6          | 76.0      | 64.3      | 63.3      |
> | **✓**     | **54.3**  | **49.8**         | **35.9**        | **14.7**               | **48.8**        | **29.9**             | **73.2** | **73.3**    | **56.1**      | **84.0**  | **65.9**  | **65.5**  |
>
> ---
>
> In addition to Table 6, we conducted an ablation study where the stage 2 and 3 are combined. Specifically, we run PLM 3B with a combined data blend (total 90M+) at 36 max number of tiles and 32 max number of frames setting (i.e., stage 3 setup) and compare with PLM 3B below:
>
> ### **Image Benchmarks**
> |              |   AVG        |   DocVQA     |   ChartQA    |   TextVQA    |   Info.QA   |   AI2D       |   OCRBench   |   MMMU       |   VQAv2      |   OK-VQA     |   VizWiz   |   SEED       |   BLINK      |   CVBench    |   RealworldQA   |   VSR        |   POPE       |
> | :----------: | :----------: | :----------: | :----------: | :----------: | :----------------------------: | :----------: | :----------: | :----------: | :----------: | :----------: | :--------: | :----------: | :----------: | :----------: | :-------------: | :----------: | :----------: |
> |   Stage2+3 (combined)   |   75.3       |   91.8       |   82.6       |   **84.5**   |   71.2                         |   90.0         |   **83.7**   |   39.9       |   83.6       |   **67.4**   |   62.8     |   78.5       |   54.6       |   72.5       |   **72.8**      |   79.9       |   **89.6**   |
> |   PLM-3B      |   **76.5**   |   **93.8**   |   **84.3**   |   84.3       |   **74.6**                     |   **90.9**   |   83.0         |   **41.2**   |   **84.3**   |   66.8       |   **64**   |   **78.5**   |   **55.4**   |   **81.4**   |   72.4          |   **80.4**   |   88.7       |
>
>
>
> ### **Video benchmarks**
> |     | AVG | MVBench | NextQA | Percept. Test | STAR | VideoMME | ActivityNetQA | EgoSchema (val) | Temp. Bench | TOMATO | MotionBench | TempCompass | CG-Bench | Charades| VideoHall. | EventHall. |
> |:---:|:---:|:---:|:---:|:---:|:---:|:---:|:---:|:---:|:---:|:---:|:---:|:---:|:---:|:---:|:---:|:---:|
> | Stage2+3 (combined) | 60.5 | 71.6 | **84.4** | 73.9 | 82.9 | **56.7** | **67.5** | 71.8 | **25.7** | 28.5 | 57.2 | **69.8** | 44.9 | 47 | 53.4 | 71.9 |
> | PLM3B | **62.5** | 74.7 | 83.4 | **79.3** | **84.8** | 54.9 | 66.2 | **72.6** | 23.4 | **30.9** | **60.4** | 69.3 | **47.2** | **57.7** | **55.5** | **76.5** |
>
> Despite nearly doubling the total training FLOPs, the combined setup results in overall worse performance compared to our three-stage training pipeline (on average **1.2** image, **2.0** video). This highlights the importance of the staged design where:
> - Stage 1 warms up the projector (MLP) with small image-caption data.
> - Stage 2 enables broad scaling using synthetic image and video data at moderate input resolution.
> - Stage 3 fine-tunes the model on high-quality human-labeled data for high input resolution.
>
> This progressive approach helps stabilize training and distribute training FLOPs efficiently. We note that training MLLMs in multiple stages is now becoming a standard practice as seen in previous works such as LLaVA-OneVision [1], Qwen-VL [2], Molmo [3], and InternVL-2 [4], etc.
>
> ---
>
> [1] LLaVA-OneVision: Easy Visual Task Transfer [Li et al., arXiv 2024]
>
> [2] Qwen-VL: A Versatile Vision-Language Model for Understanding, Localization, Text Reading, and Beyond [Bai et al., arXiv 2023]
>
> [3] Molmo and Pixmo: Open Weights and Open Data for State-of-the-Art Multimodal Models [Deitke et al., arXiv 2024]
>
> [4] Expanding performance boundaries of open-source multimodal models with model, data, and test-time scaling [Chen et al., arXiv 2025]
>
> ---
>
> > 2. Will other synthetic data (66.1M) besides the high-quality human-annotated video data also be open-sourced?
>
> Yes, all datasets used in this work, including both the 66.1M synthetic data and the 2.8M human-annotated video data, will be open-sourced to support transparency and reproducibility in future research.

---

### Official Review · Reviewer_dG2e · 2025-07-03

**Clarity:** 4
**Significance:** 3
**Originality:** 3
**Rating:** 5
**Confidence:** 4

**Summary:**

The paper introduces the Perception Language Model (PLM), an open and fully reproducible model aimed at advancing visual and video understanding. The authors address the limitations of proprietary vision-language models by providing transparent training processes, including both synthetic and human-annotated data. They also introduce PLM-VideoBench, a new benchmark suite focused on fine-grained video understanding and spatio-temporal reasoning. Through this work, the authors aim to enhance reproducibility in VLM research and to bridge gaps in detailed video understanding, especially in tasks requiring fine-grained temporal and spatial reasoning.

**Questions:**

Can the author provide a plot for the distribution of synthetic data and real annotated data. This may be helpful to analyze the limitation of synthetic data in section 3.

**Ethical Concerns:**

["NO or VERY MINOR ethics concerns only"]

**Final Justification:**

After reviewing your response and the other reviewers’ comments, I’m maintaining my recommendation for acceptance.

**Limitations:**

The author can add potential negative societal impact, especially training with unverified synthetic data.

**Quality:**

4

**Strengths And Weaknesses:**

**Strength**

1) The authors emphasize transparency and reproducibility, offering complete datasets, training recipes, and models to facilitate further research in visual understanding.

2)  The release of new 2.8 M human-labeled instances of fine-grained video question-answer pairs and spatio-temporally grounded captions significantly enhances the data available for training VLMs, addressing existing gaps in video understanding tasks.

3) PLM performs competitively against existing state-of-the-art models, including proprietary models, across multiple image and video benchmarks, showcasing its efficacy despite the use of only open-access and synthetic data.

4) The introduction of PLM-VideoBench is a valuable contribution for evaluating challenging video understanding tasks that involve spatial, temporal, and fine-grained activity reasoning.

**Weakness**

A deeper analysis of how the syntheic/realistic quality and proportion of synthetic data contribute to video reasoning could be valuable to the community.

---

> ### Author Rebuttal · Authors · 2025-07-30
>
> We thank the reviewer for the acknowledgment of our efforts toward transparency and reproducibility, including the release of 2.8M human-labeled video question-answer pairs and grounded captions. We also value the recognition of PLM’s competitive performance across image and video benchmarks, against both open-source and proprietary models, as well as the introduction of PLM-VideoBench for fine-grained spatio-temporal evaluation.
>
> ---
>
> > A deeper analysis of how the synthetic/realistic quality and proportion of synthetic data contribute to video reasoning could be valuable to the community.
>
> We thank the reviewer for this suggestion. A deeper analysis of synthetic data quality and proportion is indeed valuable, and we provide relevant results in Appendix B. Specifically, we show synthetic data scaling laws for video reasoning tasks such as EgoSchema, MVBench, VideoMME, and PerceptionTest, analyzing model performance with respect to both vision encoder size (Figure 7) and input size (Figure 8). Further, dataset-specific trends are available in Figure 9. We hope these analyses offer useful insights into the role of synthetic data in video understanding.
>
> ---
>
> > Can the author provide a plot for the distribution of synthetic data and real annotated data. This may be helpful to analyze the limitation of synthetic data in section 3.
>
> The distribution of synthetic and human-annotated data used in training is detailed in Table 9 of Appendix A. This table breaks down the number of samples from each dataset, covering both synthetic and manually annotated data across image and video tasks. While we are currently unable to include plots in the author response due to NeurIPS policy, we will include a visual representation of this distribution in the final version of the paper.
>
> ---
>
> > Limitations:
> > The author can add potential negative societal impact, especially training with unverified synthetic data.
>
> We acknowledge that training with large-scale synthetic data may carry risks related to data quality and unintended biases. In our Appendix M (L1636), we have discussed the broader impact of our project. Specifically, our dataset has undergone a number of careful data mitigation (CSAM, NSFW, etc.), as well as additional filtering of harmful content and human subject and face-blurring.

---

> > ### Comment · Reviewer_dG2e · 2025-08-04
> >
> > Thank you for the thorough rebuttal. My thanks also go to the other reviewers, AC and PC for their efforts. After reviewing your response and the other reviewers’ comments, I’m maintaining my recommendation for acceptance.

---

### Official Review · Reviewer_yuJr · 2025-07-15

**Clarity:** 4
**Significance:** 4
**Originality:** 3
**Rating:** 6
**Confidence:** 4

**Summary:**

This paper constructs a fully open and reproducible Perception Language Model (PLM) and releases a large-scale, high-quality, human-annotated video question-answering and spatio-temporally grounded video captioning dataset. Simultaneously, the paper introduces a new PLM-VideoBench benchmark, aiming to promote transparent research in image and video fine-grained understanding and address the limitations of existing synthetic data in handling complex tasks.

**Questions:**

1. I am curious how the model learns the ability for temporal localization (obtaining action frame indices) in RTLoc and RDCap. Apart from the training data and the positional encoding design in the Perception Encoder, are there any other factors that contribute to the model's temporal localization capability?

2. Can the PLM-VideoBench benchmark effectively evaluate the model's "long-term dependency" and "complex interaction" reasoning abilities? For example, are there specific questions designed to test the model's ability to understand the same event or person across multiple (non-contiguous) video segments?
3. Regarding Table 6's ablation study: Is the only varying parameter the data ratio, meaning the total SFT data, including PLM data, is also capped at 4M? If not, and the experiment states that the additional introduction of PLM data brings performance gains, it would weaken the effectiveness of PLM-built data.

**Ethical Concerns:**

["NO or VERY MINOR ethics concerns only"]

**Final Justification:**

My concerns have been well addressed; I raise my score to Strong Accept.

**Limitations:**

yes

**Paper Formatting Concerns:**

No major formatting issues have been found

**Quality:**

4

**Strengths And Weaknesses:**

**Strengths**

1. **Complete Data Construction Plan**: The paper provides a complete and comprehensive solution for constructing VLM image/video understanding data based on open-source information. This solution thoroughly discusses the advantages and disadvantages of synthetic and human-annotated data, enlightening practitioners on how to purposefully build visual understanding data.

2. **Adapted Open-Source Training Strategy**: The paper offers a set of training strategies adapted to the aforementioned data plan and based on open-source solutions. This training strategy achieves performance comparable to open-weight models without the limitations of distilling from proprietary models, pushing the performance ceiling for fully open models.

3. **More Challenging Video Benchmark**: The paper introduces a more challenging video benchmark, PLM-VideoBench. This benchmark is constructed based on human annotation, enhancing the evaluation of fine-grained and spatio-temporally grounded video understanding, thus complementing existing video benchmarks.

**Weaknesses**

1. **Insufficient Discussion on Resource Consumption**: The paper emphasizes reproducibility as a contribution but does not detail the specific computational resources required for training and deploying PLM (e.g., number of GPUs, training duration, total FLOPs).

2. **Architectural Generality Rather Than Innovation**: The model architecture (vision encoder + MLP projector + language decoder) is quite common in current VLM research. While the paper's focus is on openness, data, and training recipes, it lacks significant innovation at the model architecture level.

---

> ### Author Rebuttal · Authors · 2025-07-30
>
> We thank the reviewer for recognizing the contributions of our work. In particular, the acknowledgment of the fully open and reproducible Perception Language Model, the release of large-scale, high-quality human-annotated video QA and grounded captioning data, and the introduction of the PLM-VideoBench benchmark for fine-grained video understanding.
>
> Below, we provide answers to the reviewer's questions.
>
> ---
> > 1. **Insufficient Discussion on Resource Consumption**: The paper emphasizes reproducibility as a contribution but does not detail the specific computational resources required for training and deploying PLM (e.g., number of GPUs, training duration, total FLOPs).
>
> The full training process for the PLM-3B model, including all three stages, takes approximately 5.5 days on 128 H100 GPUs. The PLM-1B model requires less training time, while the PLM-8B model takes slightly longer. The detailed breakdown is listed in the table below.
>
> | Model              | Stage 1 (GPUs & Time) | Stage 2 (GPUs & Time) | Stage 3 (GPUs & Time) |
> |-------------------|------------------------|------------------------|------------------------|
> | PLM-1B (PE L/14)  | 8 GPUs & 3 hours       | 128 GPUs & 1.5 Days    | 128 GPUs & 2.0 Days    |
> | PLM-3B (PE L/14)  | 8 GPUs & 4 hours       | 128 GPUs & 3.0 Days    | 128 GPUs & 2.5 Days    |
> | PLM-8B (PE G/14)  | 8 GPUs & 6 hours       | 256 GPUs & 3.0 Days    | 256 GPUs & 2.5 Days    |
>
> Further, at inference time, all models can run efficiently on a single GPU: the 1B model on a 24GB GPU, the 3B model on a 40GB GPU, and the 8B model on an 80GB GPU. We will include these details in the final version to support reproducibility.
>
> ---
>
> > 2. **Architectural Generality Rather Than Innovation**: The model architecture (vision encoder + MLP projector + language decoder) is quite common in current VLM research. While the paper's focus is on openness, data, and training recipes, it lacks significant innovation at the model architecture level.
>
> We agree with the reviewer that the model architecture follows commonly used components (vision encoder + MLP projector + language decoder) and architectural innovation is not the focus of this work. Instead, we focus on developing a reproducible recipe for a broad spectrum of VLM models. Therefore, we chose to keep the architecture simple with the most popular choice of VLM components, as also in LLaVA [1], QwenVL [2], Molmo [3] and InternVL [4].
>
> Our primary contributions lie in building a fully open and reproducible pipeline, including a scalable synthetic data engine, analysis of scaling behavior across data modalities, and identifying critical gaps in existing training data. To address these, we introduce the PLM-FGQA and PLM-STC (our human annotated datasets) and the PLM-VideoBench benchmark, along with an open-source training recipe. Together, these components result in a fully open model that achieves state-of-the-art performance across 40+ benchmarks.
>
> ---
> [1] Visual instruction tuning [Liu et al., arXiv 2023]
>
> [2] Qwen-VL: A Versatile Vision-Language Model for Understanding, Localization, Text Reading, and Beyond [Bai et al., arXiv 2023]
>
> [3] Molmo and Pixmo: Open Weights and Open Data for State-of-the-Art Multimodal Models [Deitke et al., arXiv 2024]
>
> [4] Internvl: Scaling up vision foundation models and aligning for generic visual-linguistic tasks [Chen et al., arXiv 2024]
>
> ---
>
> > 1. I am curious how the model learns the ability for temporal localization (obtaining action frame indices) in RTLoc and RDCap. Apart from the training data and the positional encoding design in the Perception Encoder, are there any other factors that contribute to the model's temporal localization capability?
>
> In addition to the training data and the temporal positional encoding in the Perception Encoder [1], the way we prompt the model also plays a key role in enabling temporal localization. Specifically, we include the number of sampled frames in the video as part of the input prompt, which helps the model frame the task as predicting the start and end frame indices within a known temporal window. We find that this explicit prompting improves the model's ability to learn temporal boundaries in tasks such as RTLoc and RDCap.
>
> ---
> [1] Perception Encoder: The best visual embeddings are not at the output of the network [Bolya et al., arXiv 2025]
>
> ---
>
> > 2. Can the PLM-VideoBench benchmark effectively evaluate the model's "long-term dependency" and "complex interaction" reasoning abilities? For example, are there specific questions designed to test the model's ability to understand the same event or person across multiple (non-contiguous) video segments?
>
> Yes, PLM-VideoBench is designed to evaluate long-term dependency and complex interaction reasoning abilities. In particular, the RDCap task includes annotations for each subject in the form of multiple non-contiguous and non-overlapping (start, end, action-focused caption) triplets. These triplets describe distinct temporally grounded actions performed by the same subject across different segments of the video. Further, RDCap explicitly provides temporal boundaries where the subject is not visible in the frame. During evaluation, the model is expected to generate these multiple event descriptions grounded in time, which directly tests its ability to track entities and reason over extended temporal contexts.
>
> ---
>
> > 3. Regarding Table 6's ablation study: Is the only varying parameter the data ratio, meaning the total SFT data, including PLM data, is also capped at 4M? If not, and the experiment states that the additional introduction of PLM data brings performance gains, it would weaken the effectiveness of PLM-built data.
>
> Thank you for your insightful question regarding the ablation study in Table 6. We would like to argue that controlling the number of total data samples for each ablation blend introduces greater confounding factors, since our SFT blend (stage 3) is a huge collection from different data sources. Instead, we maintain the standard practice of training all settings using 1 epoch in total, ensuring a consistent evaluation framework.

---

### Decision · Program_Chairs · 2025-09-17

**Decision:**

Accept (spotlight)

**Comment:**

All reviewers recommend clear acceptance, highlighting the paper’s solid technical contributions, strong empirical validation, and clear impact on the vision-language modeling community. The open-source release further strengthens its significance and potential influence.

Several concerns raised by the reviewers were adequately addressed in the rebuttal, including:
- Discussion on resource consumption (reviewer `yuJr`)
- Innovation lack at the model architecture level (reviewer `yuJr`)
- Discussion on synthetic data (reviewer `dG2e`, `e1P7`)
- Unclear significance of each training stage (reviewer `s99d`)

Given the uniformly positive reviews and its contributions to both research and practice, I recommend acceptance.